# Meta-analysis indicates better climate adaptation and mitigation performance of hybrid engineering-natural coastal defence measures

Lam Thi Mai Huynh ©[1] ✉, Jie Su ©[2] ✉, Quanli Wang[2], Lindsay C. Stringer ©[3,4], Adam D. Switzer ©[5,6] & Alexandros Gasparatos ©[2,7] ✉

Traditional approaches to coastal defence often struggle to reduce the risks of accelerated climate change. Incorporating nature-based components into coastal defences may enhance adaptation to climate change with added benefits, but we need to compare their performance against conventional hard measures. We conduct a meta-analysis that compares the performances of hard, hybrid, soft and natural measures for coastal defence across different functions of risk reduction, climate change mitigation, and cost-effectiveness. Hybrid and soft measures offer higher risk reduction and climate change mitigation benefits than unvegetated natural systems, while performing on par with natural measures. Soft and hybrid measures are more cost-effective than hard measures, while hybrid measures provide the highest hazard reduction among all measures. All coastal defence measures have a positive economic return over a 20-year period. Mindful of risk context, our results provide strong an evidence-base for integrating and upscaling nature-based components into coastal defences in lower risk areas.

Coastal areas contain about 40% of the world's population and three-quarters of the large cities[1]. Climate change-induced disasters from sea-level rise, increased climate variability, and more frequent/intense droughts, floods, and storms, significantly threaten coastal communities globally[2,3]. The high-end risk scenario of the Intergovernmental Panel for Climate Change (IPCC) estimates increases in the magnitude and frequency of tropical cyclones and a 2-m sea-level rise by 2100[4]. This suggests a strong need for effective coastal defence in order to keep pace with accelerated climate change[5].

Traditional approaches to coastal defence include hard engineering measures such as breakwaters, dikes, dams, groins, and levees

(called hard measures for the remainder of the paper)[6]. However, their maintenance costs could be extremely high under future climate change scenarios when considering needs for continuous upgrades/repairs[7]. The annual cost globally for dikes alone could be USD 12-71 billion by 2100[8].

Coastal defence options that contain natural components have received attention as more sustainable and cost-effective measures compared to conventional hard measures[9]. These encompass very diverse natural, soft and hybrid measures that are collectively called "Nature-based solutions" (NbS). Natural measures include coastal ecosystems such as mangroves, seagrass beds, coral reefs, or

[1]Graduate Program in Sustainability Science - Global Leadership Initiative (GPSS-GLI), The University of Tokyo, Kashiwa City, Japan. [2]Institute for Future Initiatives (IFI), The University of Tokyo, Bunkyo-ku, Tokyo, Japan. [3]York Environmental Sustainability Institute, University of York, York, UK. [4]Department of Environment and Geography, University of York, York, UK. [5]Earth Observatory of Singapore, Nanyang Technological University, Singapore, Singapore. [6]Asian School of the Environment, Nanyang Technological University, Singapore, Singapore. [7]Institute for the Advanced Study of Sustainability (UNU-IAS), United Nations University, Shibuya-ku, Tokyo, Japan. ✉e-mail: lam.huynh@s.k.u-tokyo.ac.jp; jie.su.eco@outlook.com; gasparatos.alex@gmail.com

tidal marshes situated between coastal communities and the lowest tides, which protect against hazards such as sea level rise and coastal flooding from storm surges and high waves[10]. Soft measures utilise natural ecosystems and the environment to reduce coastal risks and achieve coastal defence and adaptation[10], and can entail the restoration, rehabilitation, reforestation, and plantation of such marine ecosystems and/or beach nourishment and sand dune planting. Hybrid measures combine hard engineering structures and soft measures (e.g. breakwaters in front of saltmarshes or mangroves), essentially offering hybrid engineering-natural solutions for coastal protection and adaptation[11].

It has been argued that, by virtue of their ability to self-adapt as living systems, NbS (whether natural, soft, or hybrid measures) can be more cost-effective alternatives to hard measures for coastal defence under a changing climate[12,13]. Furthermore, adopting NbS for coastal defence can conserve and restore natural habitats that provide multiple ecosystem services, including food and carbon sequestration[14]. Beyond climate change adaptation, some NbS for coastal defence can contribute to climate change mitigation and human wellbeing. However, their wider adoption is still limited, and not often advocated as a standard approach to coastal defence[11].

Optimum approaches to coastal defence and adaptation depend on local context-specific factors. Importantly, optimum approacher are arguably unlikely to rely exclusively on hard, hybrid, soft, or natural approaches[7,15], but will likely consist of rather diverse portfolios of options that carefully consider risk urgency, risk intensity, and the local context[15]. Robust global comparisons about the performance of hard, hybrid, soft, and natural measures for coastal defence and climate change adaptation (and the influencing factors) are particularly needed. Such comparative knowledge can inform decision-making for coastal infrastructures, facilitate the sharing of best practices, and provide guidelines for building sustainable and resilient coastal communities[7].

The current global evidence about the comparative performance of natural, soft, hybrid, and hard measures for coastal defence and adaptation in the English language literature is rather fragmented. First, to date, most research comparing their performance is usually restricted to specific study areas, functions (e.g. wave attenuation[16], shoreline stabilisation[17]) and/or the underlying costs/benefits[18]. Second, there is generally clear evidence about the effectiveness of hard measures for coastal defence[19,20], but less so for NbS. Third, the rapid loss globally of natural habitats with adaptation potential creates opportunities for ecosystem restoration via hybrid and soft measures[21,22], but there are knowledge gaps about the performance of restored habitats as part of soft and hybrid measures for risk reduction and climate change mitigation. Systematic syntheses of the performance of soft and hybrid measures for coastal defence is limited, as is the evidence of how such measures perform compared to natural habitats and conventional hard measures. Most previous systematic reviews of coastal defence options relied on the narrative-based meta-synthesis of empirical studies[10,11] Though valuable, such studies are also limited by their inability to deal with the statistical variation in outcomes between studies[23]. Conversely, quantitative meta-analyses can be informative and robust when comparing outcomes from multiple studies[23].

Here, we present a global multi-dimensional meta-analysis that compares the performance of four coastal defence options (hard, hybrid, soft, and natural measures) for coastal adaptation. It utilises insights mainly from the peer-reviewed literature and secondarily from grey literature. We focus on three performance dimensions, namely risk reduction, climate change mitigation, and cost-effectiveness. First, we compare the risk reduction and mitigation performances of coastal defences that entail human interventions (i.e. hard, hybrid, soft measures) with two comparative bases (natural measures and unvegetated natural systems) Second, we compare the performance between hybrid, soft, and hard measures Third, we examine the effect of different ecosystem types and baseline level of risks on the performance of NbS; Fourth, we quantify the costs and benefits associated with these coastal defence measures over a 20-year period. Our results provide an evidence base to guide decision-making for coastal defence and climate change adaptation, with important implications for policy, practice and future research.

## Results
### General literature patterns
We identified 304 studies assessing the effectiveness and performance of coastal defence options, with 39% ($N = 119$) reporting risk reduction, 24.7% ($N = 75$) reporting climate change mitigation, and 36.3 % ($N = 110$) reporting costs/benefits. Supplementary Fig. 7 (Supplementary Materials) shows the geographical distribution of the reviewed studies. The studies span 55 countries and territories, in North America (36.6%), Asia (35.5%), Europe (13.4%), Oceania (10.7%), Africa (2.3%) and Central/South America (1.5%). Collectively, the studies contain 875 observations about the effectiveness of coastal defence options, including 585 observations (66.9%) on soft measures, 187 observations (21.4%) on hybrid measures, and 103 observations (11.8%) on hard measures (see Supplementary Data 1).

### Group performance meta-analysis
The meta-analysis compared the performance of coastal defence options that entail human interventions (i.e soft, hybrid, hard measures) with two comparative bases that lack conscious human effort towards adaptation (i.e. natural measures, unvegetated natural systems). We compared physical performance across two dimensions, namely risk reduction and climate change mitigation. Risk reduction encompasses the functions of wave attenuation (at high and low wave energy level), shoreline stabilisation, accretion change, elevation change, and sediment accumulation, while climate change mitigation functions include carbon storage and greenhouse gas (GHG) emissions (see Methods).

Compared to natural measures (e.g. natural saltmarshes, mangroves, coral reefs, seagrass beds), soft, hybrid, and hard measures have similar overall performance, although several notable differences were observed for individual functions. When comparing "soft vs. natural", 236 observations from 72 studies (Fig. 1a) suggest that soft measures are on aggregate much more effective in risk reduction (Standard Mean Difference SMD = 1.73, 95% Confidence Intervals 95% CIs = 0.13–3.34, number of observation $n = 61$), particularly for accretion (SMD = 2.21, 95% CIs=0.17–4.25, $n = 26$) and elevation change (SMD = 2.53, 95%CIs=0.31–4.74, $n = 19$). Regarding climate change mitigation, the levels of carbon storage (SMD = −0.13, 95% CIs = −0.89–0.63, $n = 100$) and GHG emissions (SMD = −0.03, 95% CIs = −0.94–0.89, $n = 74$) do not differ substantially between soft and natural measures.

When comparing "hybrid vs. natural", 38 observations from 18 studies (Fig. 1b) indicate that hybrid measures exhibit similar overall performance compared to natural measures for risk reduction functions (SMD = 1.22, 95%CIs = −1.07–3.51, $n = 29$). There were no major differences for wave attenuation at low wave energy conditions (SMD = 5.43, 95%CIs = −4.92–15.42, $n = 5$), elevation change (SMD = −0.15, 95%CIs = −3.84–3.55, $n = 13$), sediment accumulation (SMD = 3.34, 95%CIs = −1.13–7.81, $n = 10$). However, hybrid measures are much less effective for carbon storage than natural habitats (SMD = −1.51, 95%CIs = −3.00–0.02, $n = 9$) (see Supplementary Data).

When comparing "hard vs. natural" (Fig. 1.c), these defence options exhibit similar performances in terms of overall risk reduction (SMD = −2.26, 95%Cis = −6.43–1.91, $n = 12$) and overall shoreline response (SMD = −0.03, 95%Cis = −6.43–1.91, $n = 6$). Hard measures perform worse when compared to natural measures for wave attenuation in low wave energy conditions (SMD = −0.97, 95% CIs =

−1.84 − −0.09, n = 5). However, considering the limited number of observations and studies, the findings of our meta-analysis for these group comparisons should be interpreted with caution. Climate change mitigation functions are not available for hard measures (see Methods).

Compared to unvegetated natural systems (e.g. tidal flats, bare land), hard, hybrid, and soft measures have on aggregate, a much better performance for adaptation. In total, 126 observations from 47 studies compare the performance of "soft vs. unvegetated natural systems". For risk reduction functions, soft measures perform better for elevation change (SMD = 3.70, 95% CIs = 1.05–6.34, n = 12), sediment accumulation (SMD = 1.68, 95%CIs = 0.08-3.27, n = 8), and wave

attenuation (overall SMD = 6.02, 95%CIs = 0.76–11.29, n = 25) at both high and low wave energy levels. Regarding climate change mitigation, restored habitats from soft measures are much more effective in carbon storage (SMD = 5.98, 95%CIs = 0.50–11.47, n = 38) but emit significantly higher amounts of GHGs (SMD = −1.47, 95%CIs = −2.21 to −0.72, n = 10) than unvegetated natural systems.

To compare "hybrid vs. unvegetated natural systems", the results indicate on aggregate the much better performance of hybrid measures (SMD = 5.89, 95%CIs = 2.50-9.27, n = 62) (Fig. 1e). These patterns are also visible for individual risk reduction functions such as sediment accumulation (SMD = 1.68, 95%CIs = 0.08-3.27, n = 8) and elevation change (SMD = 0.54 95%CIs = 0.33–0.75, n = 24). For wave

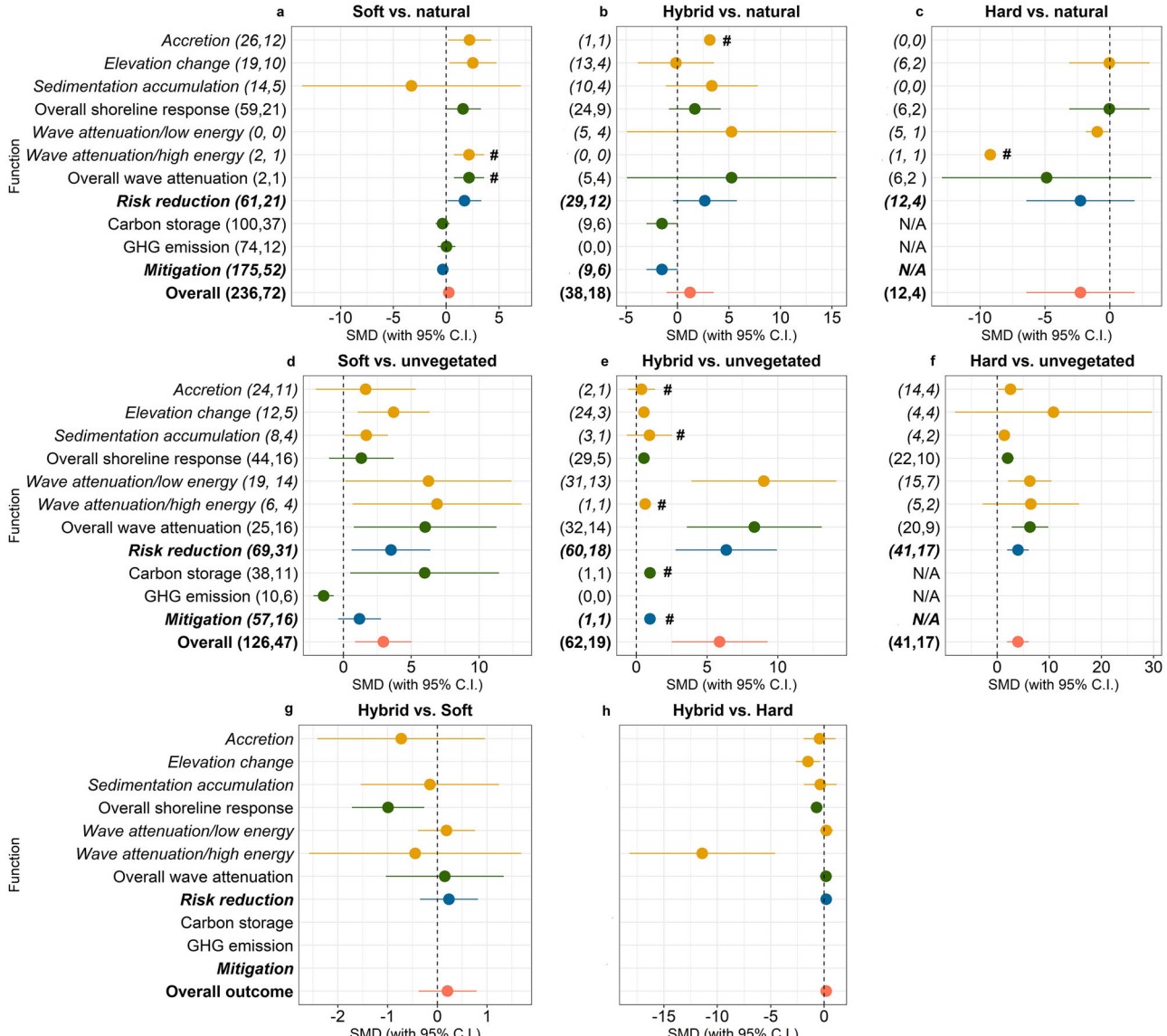

**Fig. 1 | Mean effect size for different functions. a** Soft vs. natural, **b** Hybrid vs. natural, **c** Hard vs. natural, **d** Soft vs. unvegetated natural systems, **e** Hybrid vs. unvegetated natural systems, **f** Hard vs. unvegetated natural systems, **g** Hybrid vs. soft, and **h** Hybrid vs. hard. Hedges' g was used to estimate the standardised mean difference (SMD) between two coastal defence options. If the bar falls into the positive side of the plot, we interpreted that the coastal defence option on the left of 'vs.' provides the given function at a higher level than the option on the right of 'vs.'. Conversely, if the bar falls into the negative side means the opposite. In **a–f** the first numbers in parentheses indicate the number of observations and the second numbers in parentheses indicate the number of studies included in each

calculation. Carbon storage and GHG emissions are not available for hard measures and are noted by 'N/A'. Ranges indicated with '#' in **a–f** denote functions with small number of observations that require cautious interpretation and generalisation. Due to data limitations, we followed an indirect comparison between soft, hybrid, and hard measures. In **g, h**, SMD was calculated based on sample size (number of reviewed studies), mean (estimated SMD from the previous meta-data analysis), and standard deviation between two paired groups: vs. "hybrid vs. unvegetated natural systems" vs. "soft vs unvegetated natural systems" and "hybrid vs. unvegetated natural systems" vs. "hard vs. unvegetated natural systems" (see Methods).

attenuation, hybrid measures are much more effective in low wave energy conditions (SMD = 9.01, 95%CIs = 3.89–14.13, $n$ = 31) than in high wave energy conditions (SMD = 0.63, n = 1) (note that the latter comparison is based on only one observation and should be interpreted cautiously). For climate change mitigation, we did not find any relevant studies for comparing GHG emissions. For the carbon storage function, we found only one study that indicates that restored habitats from hybrid measures have higher carbon sequestration than unvegetated natural systems (SMD = 0.96). Due to the low number of observations, this finding should be interpreted with caution.

We could only compare the performance of "hard vs. unvegetated natural systems" for risk reduction functions but not climate change mitigation functions (see Methods for explanation). Figure 1f suggests that when compared to unvegetated natural systems from 41 observations of 17 studies, hard measures provide greater risk reduction (SMD = 3.40, 95%CIs = 2.78–6.06, $n$ = 41). Similarly, the performance of hard measures is substantially better for most individual functions: accretion change (SMD = 2.55, 95%CIs = 0.12–4.97, $n$ = 14), sedimentation accumulation (SMD = 1.37, 95%CIs = 0.69–2.04, $n$ = 4), and overall shoreline response (SMD = 2.01, 95%CIs = 0.82–3.20, $n$ = 22). These results essentially confirm what we know, namely that the hard structures are designed very specifically for risk reduction and therefore perform well for these functions despite the fact that they do not perform well for other desired functions such as climate mitigation. For the wave attenuation function, hard measures perform much better in low wave energy conditions than unvegetated systems (SMD = 6.261, 95%CIs = 2.10–10.42, $n$ = 15), but performance differences are significantly reduced in higher wave energy conditions (SMD = 6.46, 95%CIs = −2.78–15.69, $n$ = 5). However, due to the small

number of observation's these comparisons must be interpreted with caution.

Overall, coastal defence options that entail human interventions (soft, hybrid, and hard measures) perform substantially better in terms of risk reduction than non-vegetated tidal flats, while they perform on-par with natural measures. However, we should point that there is a substantially higher number of pair-wise observations in low wave energy and low-risk conditions ($n$ = 213) than medium-to-high wave energy conditions ($n$ = 49). The results indicate that soft, hybrid, and hard measures are generally much more effective in low-risk contexts than high-risk contexts, when compared to both unvegetated systems and natural measures. For climate change mitigation, soft and hybrid measures in general perform worse than natural measures but much better than unvegetated natural systems. The Cochran's Q test reveals significant heterogeneity across functions.

## Subgroup analysis by ecosystem type
When compared to paired unvegetated natural systems, measures that contain restored mangrove have the best performance compared to measures containing other restored habitats (e.g. saltmarshes, coral/ oyster reefs, other wetlands) (Fig. 2a). For individual functions, hybrid measures containing saltmarshes perform the best for wave attenuation (SMD = 7.18, 95%CIs = 1.78–12.59), while soft measures containing mangroves (SMD = 4.06, 95%CIs = 0.72–7.40) and beach and sand dune nourishment (SMD = 2.87, 95%CIs = 0.25–5.49) perform the best for shoreline response. Soft measures containing saltmarshes perform the best for carbon storage (SMD = 14.07) and GHG emissions (SMD = −0.75) compared to measures containing other habitats. For all habitat types hybrid measures slightly outperform their respective soft measures in almost all risk reduction functions (Fig. 2a). However, due

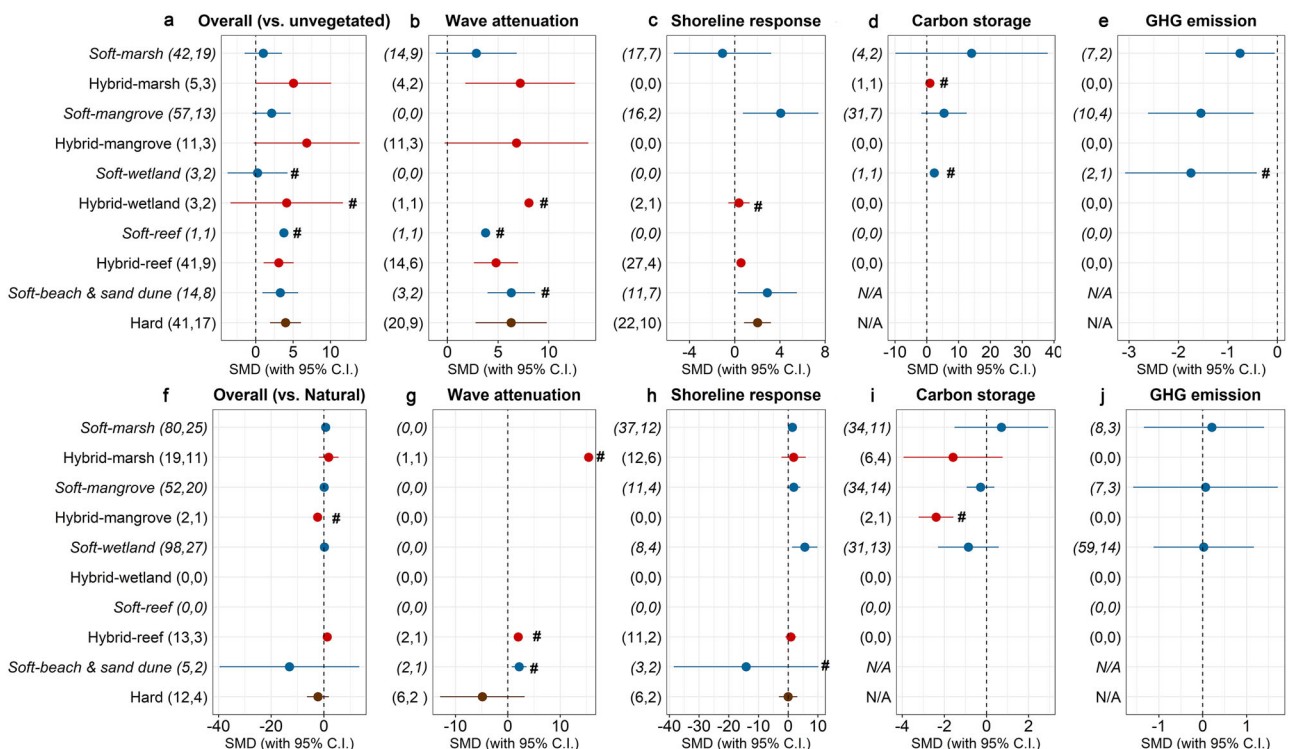

**Fig. 2 | Subgroup analysis for different ecosystem types.** Panel **a**–**e** compare functions between soft, hybrid, and hard measures with unvegetated natural systems. Panel **f**–**j** compare functions between soft, hybrid, and hard measures with natural measures. Hedges' g was used to estimate effect sizes as the difference in the means between the two groups. If the bar falls into the positive side of the plot, we interpreted that soft or hybrid measures provide the given function at a higher level than the reference base (i.e. natural measures). If the bar falls into the negative side it means the opposite. The first numbers in parentheses indicate the number of observations and the second numbers in parentheses indicate the number of studies included in each calculation. Carbon storage and GHG emissions are not available for hard measures and are noted by 'N/A'. Ranges indicated with '#' denote functions with small number of observations that require cautious interpretation and generalization.

to the limited number of studies in this subgroup, our analysis should be interpreted with caution.

## Performance comparison between soft, hybrid, and hard measures

Due to lack of data availability, it is impossible to directly compare the performance across soft, hybrid, and hard measures with the above approach. However, we compared them indirectly by examining the performance of soft, hybrid, and hard measures with unvegetated natural systems, using the results of the meta-analysis (see Methods).

By using "unvegetated natural systems" as the third comparative base, hybrid measures perform on aggregate slightly better than soft measures (SMD = 0.26, 95%CIs = −0.04−0.57) and hard measures (SMD = 0.18, 95%CIs = −0.23−0.59), although there are no major differences between "soft vs unvegetated natural systems", "hybrid vs. unvegetated natural systems", and "hard vs. unvegetated natural systems" for risk reduction, climate change mitigation, and overall adaptation performance (Fig. 1g–h).

SMDs for "hybrid vs. unvegetated natural systems" and "soft vs unvegetated natural systems" are mostly positive for risk reduction functions and negative for climate change mitigation functions, indicating that hybrid measures may perform slightly better for risk reduction but slightly worse for climate change mitigation than soft measures. When comparing "hybrid vs. unvegetated natural systems" to "hard vs. unvegetated natural systems", hard measures may perform better for elevation change (SMD = −1.51, 95%CIs = −2.65 to −0.38) and overall shoreline response (SMD = −1.01, 95%CIs = −1.65 to −0.38). At high wave energy level, hard measures are significantly more effective in wave attenuation (SMD = −11.41, 95%CIs = −18.21 to −4.60). However, the overall risk reduction is slightly higher (although not much different) for hybrid measures compared to hard measures.

A series of tests, namely sensitivity analysis using Cook's distance for outliers, regression analysis for temporal change, and publication bias analysis using the Egger test and funnel plots suggest that the results of the meta-analysis are largely robust, with some minor exceptions (see Supplementary Box 1 and Supplementary Fig. 5-7, Supplementary Material).

## Cost-benefit analysis

We calculated Benefit-Cost Ratios (BCRs) for 96 coastal defence projects that contain information on both total costs and total benefits. Specifically, this includes 55 observations of soft measures, 19 observations of hybrid measures, and 24 observations of hard measures. The costs of coastal defence measures vary significantly between different types of measures and different habitats.

Overall, for all discount rates (−2%, 4.5%, and 8%), soft, hybrid, and hard measures show considerable economic returns on investment over a 20-year period (Fig. 3). The mean BCR was highest for soft measures (mean BCRs for discount rate of −2%, 4%, 8% are 11.08, 6.40, and 4.80, respectively), followed by hybrid measures (mean BCR: 7.18, 4.20, and 3.17 respectively), hard measures (mean BCR: 6.14, 4.11, and 3.40, respectively).

When considering habitat types, soft and hybrid measures with restored mangroves offer the highest returns on investment for all discount rates (mean BCR: 22.02, 12.90, and 9.71, respectively) (Fig. 3), followed by seagrasses (mean BCR: 9.28, 5.40, and 4.09 respectively) and salt marshes and other wetlands (mean BCR: 6.20, 3.70, and 2.10 respectively). For restored coral reefs we observe the greatest benefits in terms of natural capital but also low BCRs due to the substantially higher restoration costs (mean BCR: 3.22, 1.3, and 0.93 respectively). Soft and hybrid measures have mean BCRs >1 for all types of habitats and for all discount rates, with the exception of coral reef at 8% discount rate (mean BCR = 0.93).

For the soft and hard engineering measures, beach and sand dune nourishments, in general, have lower initial investment costs compared to other hard structures such as groynes, dikes, revetments, breakwaters, and seawalls. However, over a 20-year period, the BCRs suggest relatively similar return on investment between sand nourishment (mean BCR: 2.93, 2.00, and 1.63 respectively) to other hard structures such as dikes and breakwaters (mean BCR: 2.72, 1.90, and 1.50, respectively).

## Discussion

Figure 4 summarises the main patterns for natural, soft, hybrid, and hard measures for coastal defence. Soft, hybrid, and hard measures perform substantially better than unvegetated natural systems in terms of risk reduction and climate change mitigation functions. Performance varies between options in terms of magnitude and direction. According to Fig. 1d–f, compared to unvegetated tidal flats, soft, hybrid, and hard measures much better: (a) attenuate wave energy/height through friction and change in water depth[16,24], (b) reduce coastal erosion/flooding[10,13,25], (c) accumulate sediment[17,26], and (d) stabilise shorelines[17,25,27]. This is consistent with previous studies[22,25]. Our results also suggest that all soft, hybrid, and hard measures perform effectively risk reduction functions at low-energy or low risk conditions, but their performances decline in high-energy and high-risk conditions. In addition, soft and hybrid measures using restored coastal habitats perform much better for carbon storage, but also have substantially higher GHG emissions compared to unvegetated natural systems. Overall, the restored coastal habitats used in most reviewed NbS tend to be carbon sinks[14,26], therefore contributing to climate change mitigation.

Conversely, soft, hybrid, and hard measures perform similarly to natural measures for risk reduction and overall adaptation outcomes, but worse for climate change mitigation. The latter is influenced by ecosystem stand age[28], meaning that restored and newly developed coastal habitats may require substantial time before performing at the same level for such functions compared to mature natural habitats (i.e. natural measures)[29,30]. Soft and hybrid measures that contain some engineered component tend to cause initial seaward shift and leverage

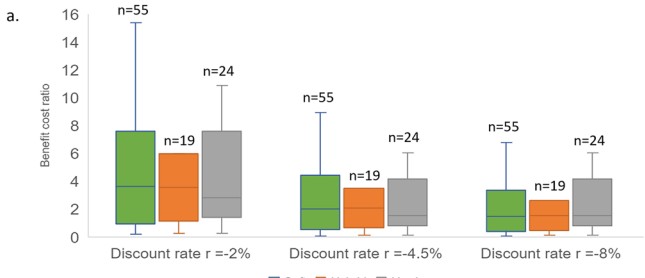

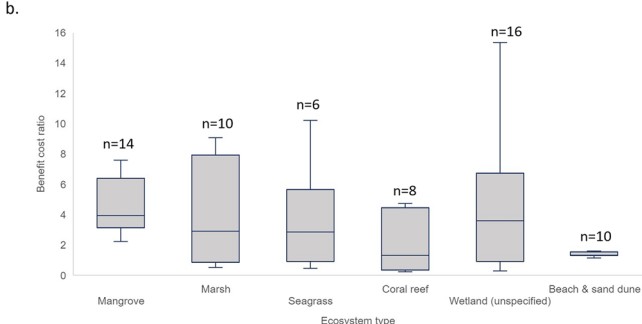

**Fig. 3 | Benefit-Cost Ratios (BCRs) of coastal defence projects.** Panel **a** illustrates BCRs of soft, hybrid, and hard coastal defence projects at discount rates of −2%, 4.5%, and 8%. Panel **b** shows BCRs for different subgroups including mangrove, marsh, seagrass, coral reef, unspecified wetland, beach, and sand dune at the discount rate −2%. Plot boxes show the minimum, first quartile, median, third quartile and maximum value. Outliers are removed in the boxes.

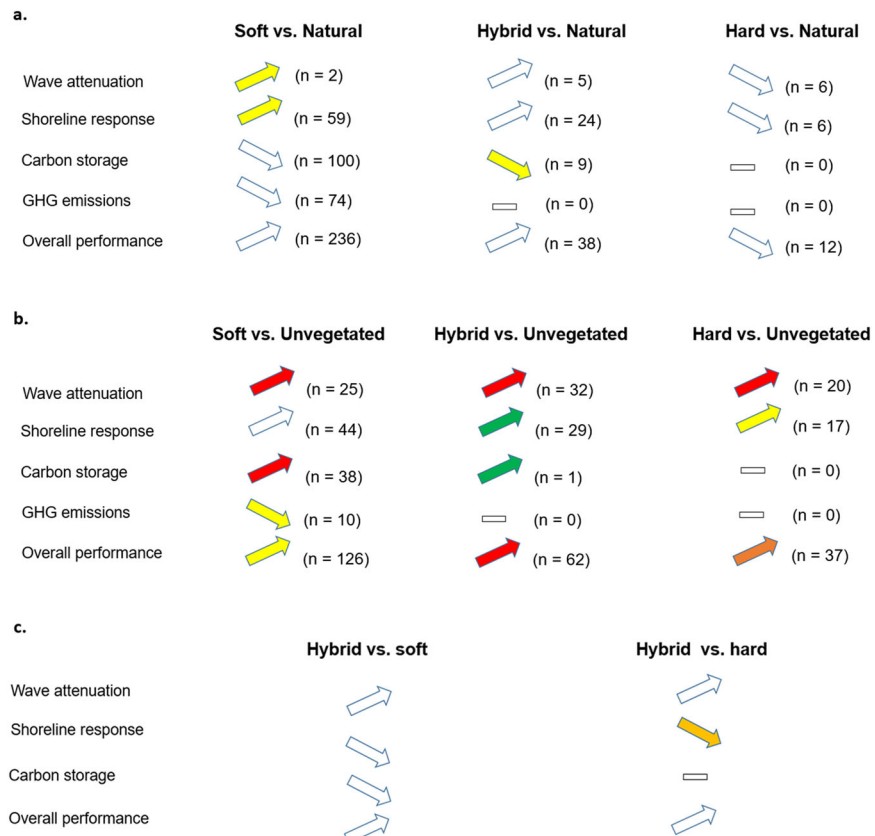

**Fig. 4 | Performance summary of the different coastal defence options.** Panel **a** compares performance between soft, hybrid, and hard measures with natural measures. Panel **b** compares performance between soft, hybrid, and hard measures with unvegetated natural systems. Panel **c** compares performance of hybrid measures with soft and hard measures. Each arrow visually represents all observations elicited from the literature for a given comparison and function, and analysed in Fig. 1. The numbers in parentheses indicate the number of observations. The upward and downward direction of the arrow denote respectively higher and lower performance of the coastal defence option on the left of 'vs.' compared to the option on the right of 'vs.' Colours denote the range of effect size. Green: small effect (absolute value: 0–1), yellow: medium effect (absolute value: 1–3), orange: large effect (absolute value: 3–5), and red: very large effect (absolute value > 5). The above colours indicate that the respective 95% CIs do not intersect with zero. The white colour indicates there is no major difference between the two respective coastal defence options (i.e. the respective 95% CIs intersect with zero). The cross bar indicates no applicable data. Arrows indicated with '#' denote aggregate functions with a small number of observations that require cautious interpretation and generalisation. Figure ideas are adapted from: Su, J., Friess, D. A. & Gasparatos, A. A meta-analysis of the ecological and economic outcomes of mangrove restoration. *Nat. Commun.* **12**, (2021), which is licensed under a Creative Commons Attribution 4.0 International License: https://creativecommons.org/licenses/by/4.0/.

ecogeomorphic feedbacks between vegetation, sediment accumulation, and organic matter accretion, therefore immediately performing better for shoreline stabilisation than natural measures[31,32]. Most studies report positive shoreline responses within relatively short periods (0–3 years) post-implementation[32–34], but the elevation and accretion differences between restored and natural habitats largely disappear given enough sediment supply over time[32–34].

Among defence options entailing human interventions, hybrid measures have a slightly better performance for most functions than soft and hard measures, although the differences are not always substantial. Hard and hybrid measures containing grey infrastructure components perform slightly better for wave attenuation and shoreline maintenance than soft and natural measures, but there is substantial variation in risk reduction outcomes for hard measures. For elevation change, the SMD of "hard vs. unvegetated natural systems" varies greatly (95%CIs = -8.1079 to 29.7024, reflecting that hard structures such as dikes and breakwaters are in most cases immediately effective against coastal erosion[35,36], but sometimes cause erosion rather than accretion in the long-term due to the complex dynamics with natural coastal processes[37,38]. Conversely, soft and hybrid measures perform better (and have lower variability in performance) for maintaining shorelines: 95%CIs = 1.0522–6.3425 for soft measures; 95%

CIs = 0.3284–0.7536 for hybrid measures. The incorporation of green components (e.g. wetlands) into coastal defence options can essentially create "living shorelines" via the hydrological reconnection of natural habitats to the sea and sediment deposition[39,40] that could protect coasts as much as hard measures, but with lower environmental risks.

We must note that the climate change mitigation function is not applicable to the analysis of the hard measures due to the lack of literature reporting pairwise comparison of GHG emissions between hard measures and other measures. This does not imply that hard structures have zero GHG impacts, as hard coastal defence projects emit significant amounts of GHGs in every phase spanning material extraction, transport, construction, maintenance, and disposal (see Supplementary Box 2, Supplementary Materials). On the other hand, soft and hybrid measures that contain ecosystem components tend to sequester carbon, with the mitigation benefits likely to increase during the project timeline as these ecosystem components mature.

It was impossible to compare directly the performance of soft, hybrid, and hard measures due to the lack of studies testing their performance in the same locations under the same environmental conditions. We identified only four papers comparing these options under controlled conditions[41–44], which were crosschecked with the

indirect comparison, providing similar observations. This lack of studies reporting paired experiments between hard, soft and hybrid measures is a significant gap in the literature. As environmental conditions affect the performance of coastal defence measures, until there are more studies directly comparing these options, it is not possible to confidently conclude which measure performs the best.

Similarly, 106 out of the 108 reviewed studies report paired experiments for risk reduction functions during periods not characterised by extreme weather events/conditions (e.g. tropical cyclones, storm surges) and non-climatic events (e.g. tsunamis). Three papers conducted paired experiments during tropical cyclones, reporting that hybrid options such as rock sills with living shorelines performed better than traditional hard measures and natural marshes (North Carolina, Hurricane Matthew)[43], while natural habitats behind the breakwaters also recovered faster than reference natural mangrove without the breakwaters (Florida, Hurricane Irma)[45], and marshes (with or without sills) perform better for erosion reduction than bulkheads during Category 1 hurricanes[44]. However, these site experiments were not directly located exactly at the landfall point (local conditions were at Category 1 and 3) and still can be considered not high-risk contexts.

More importantly, our subgroup analysis contains a much larger set of pairwise observations in low-energy conditions (213 observations) compared to medium-to-high energy conditions (49 observations). We also notice a context-specific variation in the baseline risk level across different coastal defence measures. In particular, for soft and hybrid measures related to marine ecosystems (i.e. mangroves, marshes), site experiments mostly report relatively lower incident wave energies and wave heights (i.e. <0.4 m). Conversely, site experiments for coastal defence measures such as beaches, sand dunes and hard infrastructure often report higher energy profile conditions ( > 0.4 m). Nevertheless, despite differences in the baseline risk levels between different defence measures, the scientific literature seems to be skewed towards experimental setups in low-energy conditions.

When looking critically the above we can infer that the current peer-reviewed evidence is relatively sufficient to draw conclusions on the performance of different coastal defence options for low-risk and low-energy contexts rather than high-risk contexts where effective coastal defence options are extremely critical for the safety of local communities and economic activities. The key point here is that all types of coastal defences have yet to be adequately tested through paired experiments in circumstances of extreme events and high level of risk urgency. This lack of literature is another significant knowledge gap requiring urgent attention. Until there are many more robust paired experiments in high-energy conditions and/or during extreme weather events or geophysical events like tsunamis, we emphasise the danger of any universal assumptions about the performance of coastal defence options, whether natural, soft or hybrid measures.

Finally, we should point that due to the characteristics of the underlying literature this review may be possibly biased towards the assessment of soft and hybrid measures (NbS) in the scientific literature for reasons that are beyond the control of the authors. Although hard structures are by far the most prominent coastal defence measures to date, there are fewer assessments of the performance of these structures using paired experiments in the scientific literature: 103 observations of hard measures vs. 772 observations of soft and hybrid measures in our reviewed papers. To reduce to the extent possible this bias, we conducted an additional analysis of the performance of hard structures in the grey literature and report the results in Supplementary Box 2 (see Methods). The results show that for wave attenuation functions, the effect sizes are higher in the grey literature (SMD = 19.52, 95%CIs = 6.16–33.44, $n = 26$) compared to peer-reviewed/conference papers (SMD = 6.28, 95%CIs = 2.78–9.80, $n = 9$). Despite these differences in the magnitude of effect sizes, the pool effect sizes have the same direction of the sign (positive sign, 95%CIs do not intersect with

zero). This shows that although the effect sizes are lower in the peer-reviewed literature, they reflect similar phenomena to what has been reported in the grey literature. Nevertheless, we strongly encourage the readers to interpret our findings keeping in mind with the probable bias towards NbS due to the overwhelming focus of paired experiments on these measures compared to hard measures.

All coastal defence measures entailing human interventions (soft, hybrid, hard measures) have positive economic returns on investment over a 20-year period (Fig. 3), pointing to the strong economic case for investments in such options. Despite the large variation in BCRs between different types of defence measures and ecosystem types due to the equally large variation in reported project costs and benefits, in general, soft measures are comparatively the most cost-effective, followed by hybrid and hard measures. When considering coastal habitats, NbS containing mangroves and saltmarshes have the highest BCRs (Fig. 3), as well as the best performance for wave attenuation, shoreline stabilisation and carbon storage (Fig. 2). Such cost-benefit estimations are supported by several studies[18,22,46,47].

Additionally, most of the underlying studies have not considered important benefits such as cultural ecosystem services and other intangible benefits[18], which would make the benefit calculations increase for soft and hybrid measures. Thus, the BCRs for NbS here may underestimate the total welfare contribution of such coastal defence options. Furthermore, the project lifetime of hard infrastructure could be >30 years[6,48], but here we used for the BCR comparisons a 20-year period, as it is a standard practice for ecosystem-related projects following TEEB guidelines. Thus, the BCRs for hard and hybrid measures could also underestimate some of the economic return of these coastal defence options.

Additionally, although natural measures are not included in the cost-benefit analysis, they tend to be more cost-effective than other coastal defence options, as the costs of ecosystem protection are generally lower than restoration costs, while the benefits are generally higher. For instance, previous reviews of mangrove ecosystem service valuation studies indicate higher economic benefits from natural over restored mangroves[22,49]. Such comprehensive economic lenses should be considered alongside the dimensions of risk reduction and climate change mitigation, when establishing the economic case for choosing the most appropriate coastal defence option in a given context. However, in contexts where natural measures have been destroyed or degraded, soft or hybrid options are the next best options.

Table 1–2 summarises the Strengths, Weaknesses, Opportunities, and Threats (SWOT) of these four coastal defence options. We find that all natural, soft, hybrid, and hard measures provide risk reduction functions and have high cost-effectiveness. Although hybrid measures perform slightly better than natural, soft, and hard measures in reducing risk, this is not significantly different in statistical terms (Fig. 1). Thus, the choice of coastal defence option arguably depends on the types of risks, their intensity, and the urgency for protection. Previous studies have pointed that NbS (hybrid, soft, natural measures) require substantial establishment periods, which makes them suitable only in contexts characterised by low-to-medium risk urgency[7]. For example, restored mangroves require 5-10 years[50] and restored saltmarshes around 6 years[51] to reach maturity, and thus provide the full extent of their risk reduction and climate change mitigation functions. When risk intensity is high, maintaining natural ecosystems might become burdensome, given the time requirements of natural recovery processes. At the same time, the growth of coastal habitats depends largely on the prevailing environmental conditions. These suggest that planting new habitats via soft measures may not always be appropriate in high-risk zones or in areas that cannot ecologically support these habitats.

By contrast, hard measures can be built within a relatively short timeframe and have immediate effects, particularly in contexts characterised by high risks and/or extreme conditions, where other

**Table. 1 | Strengths and weaknesses of different coastal defence options**

| | Hard measures | Natural measures | Soft measures | Hybrid measures |
|---|---|---|---|---|
| **Strengths** | - Long experience in design and implementation[84].<br>- Substantial evidence base about the types, functions, protection mechanisms and capacity, and potential risks of different structures[11].<br>- Immediate effectiveness after construction (MA), which makes them suitable for high-risk contexts[7].<br>- Possible suitability for areas with limited space[13].<br>- Positive economic return on investment (MA) | - Possible self-adaptiveness to changing climate and self-recovery after a disaster[9].<br>- Provision of multiple co-benefits, including climate change mitigation and risk reduction (MA), human wellbeing and biodiversity conservation[9].<br>- No investment costs and very high economic benefits[22]. | - Possible self-adaptiveness to changing climate and self-recovery after a disaster[9].<br>- Provision of multiple co-benefits, including climate change mitigation and risk reduction (MA), human well-being and biodiversity conservation[9].<br>- Restored habitats are more effective in risk reduction with time[7].<br>- Positive economic return on investment (MA) | - Flexibility in combining hard and soft measures, which allows for innovative context-specific practices when designing coastal defence[11].<br>- Provision of multiple co-benefits, including climate change mitigation and risk reduction (MA), human well-being and biodiversity conservation[9].<br>- Greater confidence in shoreline protection by harnessing the advantages of both hard and soft measure[25].<br>- Suitable for contexts characterised by low-to-high risk urgency (MA).<br>- Positive economic return on investment (MA). |
| **Weaknesses** | - Lack of self-adaptiveness to changing climate[11].<br>- High environmental risks (e.g. structures damage natural habitats or degrade water quality)[13].<br>- Failure to provide co-benefits associated with climate change mitigation, human wellbeing and biodiversity conservation (BL).<br>- Possibility of creating perceptions of false safety to local communities[11]. | - High vulnerability to the rapid degradation of natural coastal habitats worldwide[11].<br>- Substantial recovery time following degradation by natural or human processes[7].<br>- Insufficient for high-risk coastal zones[7].<br>- Significant space requirement for implementation, which makes them possibly unsuitable for dense urban areas[11]. | - Long time requirement to reach maturity and provide risk reduction benefits equivalent to hard structures[50,51].<br>- Effectiveness is highly dependent on ecosystem type (Fig. 2) and ecological support of the local environment[59].<br>- Lack of suitability for areas that cannot support ecosystems[11].<br>- Significant space requirement for implementation, which makes them possibly unsuitable for dense urban areas[11].<br>- Distinct human-nature interactions during implementation[59]. | - Limited implementation due to knowledge, expertise, and experiences globally (BL).<br>- Substantial effort requirement from different stakeholders (e.g. academics, policy-makers, private sector, local communities) to design the best hybrid measure in a given context[7].<br>- Negative environmental impacts of the grey infrastructure component[7].<br>- High uncertainty over operational parameters and effectiveness during implementation[7].<br>- Distinct human-nature interactions during implementation[59]. |

All statements are derived through expert judgement of the Authors and are based on different components of the systematic review. Statements derived from the meta-analysis are identified as (MA); Statements based on qualitative/quantitative findings or discussion points reported in individual studies are cited with a study; Statements based on the authors' critical understanding of the broader literature are indicated as (BL).

options are not feasible[7,11]. Our results indicate that hybrid options can harness the strengths and overcome some of the limitations of natural, soft and hard measures. These findings are also in line with the previous global studies[7,52]. Arguably, hybrid measures can be suitable in contexts characterised by medium-to-high risk and high urgency[7]. For example, in some hybrid options, hard components such as dikes and breakwaters can provide immediate protection for an eroding coast, while the establishment of natural or restored saltmarshes can deliver long-term shoreline stabilisation with lower environmental damage[53]. Current technological and engineering advances can broaden considerably the possible range of hybrid options and improve their feasibility and suitability. Overall, our analysis suggests that natural and soft measures work well in low-risk contexts, while hybrid and hard measures are better options for medium-to-high risk contexts.

While it might have been previously enough to consider risk reduction, cost-effectiveness and risk urgency when selecting coastal defence options, the current reality of accelerated climate change complicates further such decisions presently. A recent technical report during the 27th Conference of the Parties of the United Nations Framework Convention on Climate Change Conference (UNFCCC-COP27) positions nature-based solutions with their climate mitigation functions perceived as "core" benefits, instead of "co-" or "additional" benefits[54]. Our results confirm that NbS (either hybrid, soft or natural measures) with natural and restored coastal habitats are significant carbon sinks and very effective in mitigating climate change (Fig. 1). Thus, we argue that the ability to provide climate change mitigation functions should become an integral criterion for selecting appropriate coastal defence options.

The findings have major implications for policy and practice, particularly in view of the UNFCCC-COP26, which noted the centrality of nature-based solutions for achieving the Paris Agreement. Interpreting our results at an aggregated level, there is a clear case for investing in NbS for coastal defence and climate change adaptation considering their substantial benefits for risk reduction and climate change mitigation, as well as their cost-effectiveness. Hybrid measures could theoretically reduce the weaknesses of natural, soft and hard measures, thus holding high potential for innovation and application to various contexts. However, it is crucial to generate appropriate knowledge and guidance in designing and planning NbS that fit the application contexts, particularly when considering the multiple knowledge gaps and factors affecting their effectiveness. Pragmatically, hard measures are inevitably necessary for coastal defence in high-risk contexts, especially in the short-to-medium term before gaps are filled and novel and context-relevant approaches to NbS development and implementation are developed.

Additionally, the wide-scale implementation of hybrid and soft measures for coastal defence is still challenging and not without critique. Despite the many successful trials reviewed here, there have also been negative or mixed outcomes. Lessons learned include to avoid introducing exotic species in vulnerable coastal systems on reclaimed land to prevent environmental trade-offs[55] or to ensure that degraded land restoration does not affect communal agricultural/grazing land or exacerbate pressure on surrounding areas[56]. Notably, while most soft and hybrid measures restoring existing habitats could be effective and have low environmental risks, NbS that create entirely new ecosystems are exercises in uncertainty[7].

Unlike hard measures, the effectiveness of NbS via soft, hybrid and natural measures is largely determined by the capacity of local communities and their long-standing ways of engaging with nature[57]. Arguably for highly exposed and vulnerable communities that have limited financial resources and high levels of environmental inequity, there is a higher risk of ineffectiveness of such measures (or even a waste of financial resources or maladaptation)[58]. We identified a bias in the literature towards information from the Global North and lower risk areas. About 68% of the identified and analysed cases were in

Europe, North America, and Australia, in regions facing low-to-medium risks, where coastal communities are relatively less vulnerable. Bearing in mind this critical knowledge gap and imperfect information, we point out the risk of perceiving NbS as a "silver bullet" for coastal adaptation and "normalise" their application in the Global South, particularly in contexts characterised by high risks, vulnerability, and poverty.

Considering this context-specificity, the decision-making process should be guided by the specific policy goals, priorities, local capacity, and socioecological constraints. However, while the implementation constraints of hard engineering structures are largely attributable to technological limits, those of soft, hybrid and natural measures tend to be socio-cultural and institutional (Table 2). In this sense, knowledge asymmetry can be critical to the upscaling of NbS for coastal defence. Practitioners on the ground do not normally obtain both the ecological and technical knowledge for the effective design of the ecosystem-based and hard engineering components respectively[7]. Furthermore, they are not always engaged in the distinct social interactions across project phases[59]. Arguably the design and implementation of NbS for coastal defence and adaptation would require an interdisciplinary and trans-disciplinary mindset, which would be rather distinct to that of the design and implementation of hard measures. It might be increasingly necessary to empower the people on the ground, equip them with broader integrated knowledge of ecology and engineering, promote public acceptance, and prepare for societal shifts for effective participation and collaboration in coastal NbS design, implementation, and management.

## Methods

### Research approach and key concepts

This paper conducts a meta-analysis of the academic literature to systematically assess the performance of different types of coastal defence options to climate change-related hazards. We focus on hard, hybrid, soft, and natural measures, which depending on their type they can be located inside the sea or between communities and low tides, and provide protection against diverse climate-related natural hazards.

Within the scope of this paper, hard engineering measures (called hard measures hereafter) are defined as the coastal defence options that utilise structures such as seawalls, dikes, breakwaters, and levees[6]. Soft measures are defined as coastal defence options that utilise the natural environment to reduce coastal risks. Soft measures include (a) coastal defence options that rely on restored, rehabilitated, reforested, planted, protected, and/or managed natural habitats such as mangroves, salt marshes, coral reefs, seagrass and kelp beds or (b) coastal defence options that utilise natural systems such as beach and sand dune nourishment, dredging, and beach scraping[7]. Hybrid measures are defined as the coastal defence options that utilise a combination of hard engineering structures and soft measures, e.g. breakwaters in front of salt marshes or rock sills with oyster reefs[11]. In this category we also include environmentally-friendly engineering solutions such as vegetated grey structures. Natural measures are defined as the coastal defence options that rely on undisturbed, naturally regenerated, or degraded natural habitats[10]. Hybrid, soft, and natural measures fall under the umbrella term of Nature-based solutions (NbS). For comparative purposes we also consider unvegetated natural systems such as unvegetated sand flats, mud flats, open beaches, bare lands, or abandoned aquaculture ponds. Supplementary Table 3 provides more detailed definitions.

Among these five types, the hard, hybrid, and soft measures entail active human interventions for coastal defence. Natural measures and unvegetated natural systems with (or without) autonomous or evolutionary natural adaptation to climate change, were used as comparative bases for our meta-analysis (i.e. controls). However, we need to point out that unvegetated tidal areas can serve very different functions compared to vegetated tidal areas (e.g. open beaches for tourism). Although they may not perform well for risk reduction functions as other defence measures, they are nevertheless important landscape features.

Our systematic review compares the performance of these coastal defence measures across three dimensions: (a) risk reduction, (b) climate change mitigation, and (c) cost-effectiveness. Risk reduction refers to the ability of a coastal defence measure to reduce the intensity, frequency, or severity of losses from climate change-induced coastal disasters such as sea level rises, floods, typhoons, and erosion[3]. The risk reduction functions that fall within the scope of this review include wave attenuation (i.e. wave height reduction, wave energy reduction at low and high wave energy conditions) and shoreline response (i.e. accretion, erosion, elevation change, sedimentation accumulation). Climate change mitigation refers to the ability of the coastal defence measure to store carbon and reduce GHG emissions. Cost-effectiveness refers to the analysis of the monetary costs and benefits of the defence measires, and provides an economic lens for understanding the investment needs and the potential returns, valued in monetary terms.

The study dimensions and functions were selected based on the following criteria: (a) the dimension/function is crucial, justifiable, and appropriate in the context of coastal defence options and the types of risks being analysed; (b) the dimension/function is suitable for all five types of coastal defence options and can be assessed through consistent and established methods; (c) the techniques used to assess each dimension/function should be comparable among the comparison groups; and (d) the available information and data can enable the comparison of performance between the groups, and should be sufficient to ensure the reliability and accuracy of the meta-data analysis. The studied functions do not represent all possible risk reduction or mitigation functions the study coastal defence options. Nevertheless, the selected functions can offer a sound assessment that reflects well the currently available evidence in the academic literature.

### Literature identification and inclusion

We systematically searched the academic literature to identify quantitative studies that compared the effectiveness of different coastal defence options on different functions. We used three categories of keywords that reflected: (a) the coastal defence options (including NbS and hard engineering solutions); (b) the coastal contexts; and (c) the comparison functions (including wave attenuation, shoreline response, carbon storage, GHG emissions, and economic costs and benefits). The detailed keywords are provided in Supplementary Table 4, and were selected to include a comprehensive set of coastal defence measures in both peer-reviewed and non-reviewed studies (see below).

We applied the PRISMA principles to ensure the quality of the systematic review and meta-data analysis[60]. After downloading and removing the duplicates, we applied two filters for screening. First, the first author scanned the titles and abstracts to remove the non-relevant papers. Next, the remaining papers were read in full by the first author to determine whether they met the pre-defined selection criteria below:

(a) Studies had to be peer-reviewed or high-quality grey literature that reports empirical field data or laboratory experiments. No temporal limits were included;

(b) For the meta-data analysis: studies that examined a given function in a controlled or paired experiment between and within the two groups: i.e. (i) options entailing human interventions (i.e. soft, hybrid, and hard measures) and (ii) natural systems (i.e. natural measures and unvegetated natural systems). For inclusion, the studies must report sample size, mean, and standard variation for both controlled and paired groups on wave attenuation capacity, shoreline stabilisation, carbon storage, and GHG emissions;

**Table. 2 | Opportunities and threats for different coastal defence options**

| | Hard measures | Natural measures | Soft measures | Hybrid measures |
|---|---|---|---|---|
| **Opportunities** | - Availability of advanced engineering options to modify and develop existing hard structures for different purposes with improved performance (BL).<br>- Good financial investment outlook in the near future as hard measure are always a priority for coastal adaptation to climate change[85].<br>- High acceptability as it is the standard approach to coastal defence globally (BL). | - Widespread coastal conservation efforts globally (BL).<br>- High awareness of many local communities, policy-makers and private sector in managing and protecting natural habitats (BL).<br>- Ongoing collective international efforts and networks for protecting natural ecosystems to reverse ecosystem degradation[86].<br>- Opportunities to create collaborations with indigenous and local communities to improve community-based resource management[87].<br>- Opportunities for creating synergies to achieve climate resilience, enhance human wellbeing, and protect biodiversity[88].<br>- Alignment of policies with funding support (BL). | - Increased international visibility as 2021–2030 is the United Nation Decade on ecosystem restoration.<br>- Ongoing collective international efforts and networks for restoring natural ecosystems to reverse ecosystem degradation[86].<br>- Financial incentives to integrate NbS for coastal defence due to the high maintenance costs of hard structures (BL).<br>- Opportunities for creating synergies to achieve climate resilience, enhance human wellbeing, and protect biodiversity.[88] | - Advanced engineering can inspire innovations in the design and improve the acceptability of hybrid measures (BL).<br>- Potential to be globally accepted as a standard approach to coastal adaptation (BL).<br>- Opportunities for creating synergies to achieve climate resilience, enhance human wellbeing, and protect biodiversity (BL).<br>- Alignment of policies with funding support (BL).<br>- Potential of creating interdisciplinary and transdisciplinary approaches to normalise the application of hybrid measures[7]. |
| **Threats** | - Technological limits of the built structures[89].- Possibility of massive failures of built structures due to inappropriate design, construction, maintenance and operation[58].<br>- Financial constraints posed by funding availability[89].<br>- Institutional limits linked to inadequate governance, limited institutional capacity, lack of political will, and existing laws and procedures[89]. | - Social/cultural limits to resource management, low local capacity, and difficulty in engaging different stakeholders, education, social beliefs, and worldviews[89].<br>- Institutional limits: inadequate governance, limited institutional capacity, lack of political will, existing laws and procedures (BL).<br>- Biological limits: unsuitable environmental and ecological conditions for ecosystem growth (BL). | - Social/cultural limits to resource management limited local capacity, difficulty in integrating different stakeholders, education, social beliefs, and worldviews (BL).<br>- Institutional limits: inadequate governance, limited institutional capacity, lack of political wills, existing laws and procedures (BL).<br>- Biological limits: unsuitable environmental and ecological conditions for ecosystem growth (BL). | - Technological limits of the built structures[7].<br>- Improper design, construction, maintenance and operation can lead to massive failures of built structures (BL).- Financial limits of funding (BL).<br>- Social/cultural limits to resource management limited local capacity, difficulty in integrating different stakeholders, education, social beliefs, and worldviews (BL).<br>- Institutional limits: inadequate governance, limited institutional capacity, lack of political wills, existing laws and procedures (BL).<br>- Biological limits: unsuitable environmental and ecological conditions for ecosystem growth (BL). |

All statements are derived through expert judgement of the Authors and are based on different components of the systematic review. Statements derived from the meta-analysis are identified as (MA); Statements based on qualitative/quantitative findings or discussion points reported in individual studies are cited with a study; Statements based on the authors' critical understanding of the broader literature are indicated as (BL).

(c)  For the cost-benefit analysis: studies that reported the monetary values with standard and established evaluation methods;

(d)  Studies had to report active human intervention to coastal defence and adaptation to climate change or natural disaster risk reduction. Coastal interventions that were not associated with climate change or did not entail human intervention were excluded.

(e)  Studies had to include current or recent empirical observations. Historic or prehistoric observations were excluded.

The literature review covered studies published in the English language up to July 2023 (search date 22 July 2023), without restriction on publication date. We identified peer-reviewed literature in Elsevier Scopus and ISI Web of Science Core Collection using the article's title, abstract, and keywords to identify the relevant literature. We identified grey literature in the BASE database. To ensure the high quality of the meta-analysis, we only included conference proceedings and doctoral dissertations from non peer-reviewed literature that met our critical appraisal criteria (see next section). Data from consultancy reports, governmental reports, and reports to funders were also extracted and analysed, but due to their generally lower performance in the quality appraisal, we report this data in the supplementary material and not the main paper. This way it is possible to provide additional information of whether the effect sizes differ between the peer-reviewed and non peer-reviewed literature.

Overall, a total of 300 peer-reviewed studies and 4 non peer-reviewed studies were included in the main analysis and 9 non peer-reviewed studies were included in the Supplementary Materials. Supplementary Material Fig. 1 reports in detail the literature selection and screening process, and the number of excluded studies at each stage. Supplementary Data 1 includes the list of the included studies' title, authors, and related information.

### Critical appraisal of reviewed studies

It is critical to assess and evaluate the reliability of evidence at the level of the individual study to ensure the quality of the meta-analysis[61]. Here, we followed appraisal guidelines for ecosystem services and conservation studies and developed a checklist for internal validity including research aims and objectives, data collection, data analysis, results and conclusions, and design-specific aspects[61] (Supplementary Table 2, Supplementary material). Each study was then assessed against the checklist and categorised as having very strong evidence (score: >75%), strong evidence (score: 50-74%), moderate evidence (score: 25-48%), and weak evidence (score: <25%).

Overall, the critical quality appraisal indicated that 96% of the peer-reviewed studies have very strong evidence, 3% strong evidence, and only three studies had moderate and weak evidence. The average quality score across all peer-reviewed studies was 85.9%.

To ensure the high quality of the meta-data analysis, we only included peer and non peer-reviewed studies with very strong and strong evidence. Thus, we removed the 3 studies with moderate and weak evidence. The final database for data coding and extraction includes 304 studies. The critical appraisal of all reviewed studies can be found in the Supplementary Data 2.

### Meta-data analysis: data extraction and analysis

The meta-data analysis was conducted for comparisons of coastal defence options that entail human interventions (i.e. soft, hybrid, and hard measures) with two comparative bases (natural measures and unvegetated natural systems). This results in a total of six types of paired comparisons, namely: (a) soft vs. natural, (b) hybrid vs. natural, (c) hard vs. natural, (d) soft vs. unvegetated natural systems, (e) hybrid vs. unvegetated natural systems, and (f) hard vs. unvegetated natural systems.

As outlined in the research approach, we conducted the comparisons across two dimensions: (a) risk reduction (wave attenuation, shoreline stabilisation) and (b) climate change mitigation (carbon storage, GHG emissions). We only extracted observations that were paired both at the same site and in the same study. The extracted variables for each function are explained below.

For the wave attenuation function, we assessed three response variables, namely wave height reduction, wave energy reduction, and wave transmission coefficient. These variables are functionally related. The typical method used for estimating these variables in the field is to measure the incoming wave energy or wave height at wave recording stations along a shore transect encompassing a paired experiment between adaptation and non-adaptation[16,24]. Considering the large variation in the baseline level of risk across different studies, we extracted information about the morphodynamic characteristics in each study. Data on significant wave height, wave energy, and storm conditions are used as indicators to classify the contextual condition of all pair-wise observations into high wave energy and low wave energy profiles. Wave attenuation analysis was then conducted for comparing the performance of soft, hybrid, and hard measures in low wave energy and high wave energy contexts. A significant wave height of <1 m is used as the cut-off point for a low wave energy conditions[62] and >1 m for high wave energy conditions. Acknowledging the lack of commonly-agreed definitions and indicators of low/high wave energy conditions in the current literature and the inherent limitations of using wave height/wave energy as an indicator, this methodological decision does not aim to provide a standardised metric of wave energy profile beyond the context of this study.

For the shoreline stabilisation function, we assessed three response variables, namely accretion change rate, elevation change rate and sedimentation accumulation. We included vertical accretion rate and soil surface elevation data which were measured in the underlying studies through surface elevation tables and the feldspar-marker horizons technique[63]. Sedimentation and deposition were measured in the underlying studies using sediment traps, Petri dishes and filter papers[64]. When accretion, elevation, and sedimentation accumulation rates were measured for multiple time frames, we calculated the average annual rate across the experiment period.

For the carbon storage function, we assessed the capacity of the coastal defence option to absorb carbon dioxide from the atmosphere and accumulate it in biomass and the soil carbon pool[14]. We extracted multiple response variables that represent carbon uptake such as aboveground biomass, foliage pool, belowground biomass, fine root pool, stump pool, soil carbon uptake, gross primary production, and carbon sequestration rate. The carbon storage function was not available for hard measures.

For GHG emissions, we assessed the levels of $CO_2$, $CH_4$, and $N_2O$ emissions from the soil, belowground biomass, and soil-emergent plant structures[65]. Coastal natural habitats have formed a dense carbon pool, which means that they are potential GHG emission hotspots when degraded[14]. Along with the carbon storage functions of EbAs, it is also important to consider GHG emissions to identify possible trade-offs. For hard measures, GHG emissions could come from their construction, maintenance and operation.[66] However, following literature screening we could not find any studies reporting a paired experiment on the GHG emissions from hard/hybrid/soft engineering measures and restored/natural measures. Hence, this function was not available for comparisons containing soft engineering, hybrid and hard measures.

For each reviewed paper included in the meta-data analysis, we extracted the sample size, mean, and standard variation for both controlled and paired groups on the abovementioned functions (see Supplementary Data 3 – 9). Data were taken from the main text, tables, and figures. We used WebPlotDigitizer (available at https://automeris.io/WebPlotDigitizer/) to graphically extract data from

plots and figures. We also extracted data on general coastal defence characteristics including geographical location, longitude, latitude, implementation scale, actors, coastal defence types, implementation stage, ecosystem types, adaptation period, species name and species age, tidal range, and related environmental factors. In total, we identified 108 studies on risk reduction and 75 studies on climate change mitigation, reporting 492 observations of comparisons between study groups

We used standardised mean difference (SMD) Hedges' g to calculate for each function the effect size of coastal defence options that entailed human intervention and the reference natural system within the same study. The effect size is a statistical parameter used in meta-data analysis to compare the results of different studies that measure a common effect of interest, with adjustments for differences in scale among studies[23]. Hedges' g has been the most common measure of effect size in ecological meta-data analyses to estimate the effect as the difference in the means between two groups[23]. Hedges' g includes a correction factor for small sample size and is not affected by unequal sampling variance between paired groups[67]. We inverted the sign of SMD for the GHG emissions, wave height, wave energy, and wave transmission coefficient before combining them with other response variables to estimate the overall effect size, as the lower means of these variables are correlated with better functions.

To estimate the overall effect and each type of function across all studies, we used multivariate models which account for non-independence within individual studies. As there are often multiple response variables measured in individual studies (e.g. wave height reduction while calculating accretion rate and sediment accumulation in the same study), non-independence within individual studies is ubiquitous[68]. Thus it is not appropriate to use conventional models such as fixed-effect models and random-effect models, which assume independence between observed outcomes from studies[23]. We illustrated 95% confidence intervals (CIs) of the effect sizes in forest plots. When the 95% CIs did not intersect with zero in the forest plots, we interpreted that coastal defence options entailing human intervention (soft, hybrid, or hard measures) has a clear positive or negative effect on a given function compared to the reference base (natural unvegetated system or natural measures). If the bar fell into the positive side of the forest plot, we interpreted that the coastal defence option entailing human intervention provides the given function at a higher level than the reference base. We interpreted the opposite if the bar fell into the negative side. We do not use P-value or CIs as a conventional description of "statistical significance", as their misuse has been criticised in many studies[69,70]. Thus, in this study, 95% CIs denote the range of probable effect size with 95% confidence[71]. We avoid using the term "significant" when interpreting the results, and instead when the 95% CIs bar does not intersect with zero we interpret that the coastal defence provides the given function at a clear-cut higher or lower level than the reference system.

Cochran's Q statistic (Qt) was used to identify whether there is heterogeneity in the effect sizes across studies[23]. When the P-value is less than 0.05 for the Qt, there is variation among effect sizes by sampling error alone. We conducted subgroup analysis for ecosystem types to identify possible factors causing heterogeneity across studies. Effect sizes were calculated and compared for different ecosystem type (e.g. mangrove, salt marsh, coral reef, seagrass bed, unspecified wetland, and beach and sand dune) for different functions.

Due to data limitations, it was not possible to conduct the type of meta-data analysis outlined above comparing directly soft, hybrid, and hard measures. Due to the very limited number of studies reporting paired experiments between these coastal defence options within the same studies and sites, the extracted data was insufficient to conduct a proper meta-analysis. Hence, we opted for an indirect analysis, comparing the performance of "soft vs unvegetated natural systems",

"hybrid vs. unvegetated natural systems", and "hard vs. unvegetated natural systems" from the result of the meta-data analysis outlined above. Using "unvegetated natural systems" as a reference third comparative base, we estimated the difference in the performances of soft, hybrid, and hard measures on a given functions compared to the reference base as a proxy for indirect comparison between them.

We calculated SMD based on sample size (number of reviewed studies), mean (estimated SMD from the previous meta-data analysis), and standard deviation between two paired groups: vs. "hybrid vs. unvegetated natural systems" vs. "soft vs unvegetated natural systems" and "hybrid vs. unvegetated natural systems" vs. "hard vs. unvegetated natural systems". We used fixed-effects models to estimate the overall effect and the effects for each type of function. Forest plots were used to illustrate 95% CIs for the effect sizes, which were interpreted as explained above. This indirect comparison is arguably appropriate to some extent considering that all effect sizes from the different studies were calculated adjusting for differences in scales in the previous meta-data analysis. The extracted data on the direct comparisons between these coastal defence options (i.e. paired data from the same studies and at the same sites) were also used for simple descriptive analysis provided in the discussion, and were crosschecked with the results of the indirect analysis.

### Meta-data analysis: sensitivity analysis, temporal trends in effect sizes, and publication biases

To evaluate the robustness and reliability of the meta-data analysis, we conducted a series of tests to: (a) identify the effects of outliers and influential observations on the outcomes, (b) test the changes in the magnitude and direction of research findings over time; and (c) detect possible publication biases in the reviewed studies.

First, for the sensitivity analysis we used Cook's distance to identify outliers in the dataset that were worth checking for validity[72]. Possible outliers are indicated when a Cook's D for that data point is more than $4/n$, where n is the number of observations for the given functions[73]. After identifying the outliers, we excluded them and recalculated the pool effect sizes. We compared the previous results with the results that excluded the outliers to examine whether the outliers had a significant effect on the results.

Second, changes in the magnitude of the directions of the effect sizes over time have been repeatedly reported in ecological studies, which may jeopardise the stability of the conclusions drawn from the meta-analysis[23]. Such changes in the direction can be due to factors such as extreme influence of a high impact study on later research, tendency to prove a higher effect size than previous research, and selectively publishing results that outperform previous results[23]. To detect whether the temporal trends present in effect sizes, we conducted meta-regressions to examine the relationship between effect sizes and publication years.

Third, the possibility of publication biases was tested using funnel plots and Egger's regression[74]. Publication bias is defined as "whenever the dissemination of research is such that the effect sizes included in a meta-analysis generate different conclusions than those obtained if effect sizes for all the appropriate statistical tests that have been correctly conducted were included in the analysis"[23]. We followed the guidelines for testing publication biases proposed for biological meta-analysis[75] using the "metafor" package in R version 4.2.2, which is appropriate for small sample size bias corrected Hedges' g[76].

After conducting these tests, we critically evaluated the reliability and robustness of the results of the meta-data analysis. Rather than treating individual studies as discoveries of wider global truth, meta-data analysis synthesises the empirical information in the academic landscape. Throughout the text, we noted which results to interpret with careful consideration and the possible biases posed by the quality of the reviewed studies, and reflected further on these issues by suggesting directions for future empirical studies in the discussion.

## Cost-benefit analysis

We conducted a meta cost-benefit analysis for soft, hybrid, and hard coastal defence projects globally. We elicited costs and benefits from the peer-reviewed literature for 96 coastal defence projects, including 55 observations for soft measures, 19 observations for hybrid measures, and 24 observations for hard measures. Some studies reported the total economic costs and benefits in the same sites for the same project, while others only included individual categories of costs and benefits subject to the thematic focus of each study. To ensure the consistency and comprehensiveness of the cost-benefit analysis (and avoid double counting), we used only the total economic costs and total economic benefits in our BCR calculations.

Among the 96 projects, 50 projects reported pair-wise information on the total costs and total benefits in the same sites within the same projects, while 46 projects (all soft and hybrid projects) only reported the costs. For the missing information of the total benefits of these 46 projects, we used the value transfer approach to pair benefit values with the cost values. For this value transfer exercise we find the best match of the benefits through the Ecosystem Service Value Database (ESVD), developed by The Economics of Ecosystems and Biodiversity (TEEB) Foundation[77]. This database includes 9500 observations of monetary benefits of ecosystem services of restoration projects from over 1100 publications over various case study locations. We followed the standard value transfer approach[78,79], where the benefit was identified in the ESVD database that best matches each coastal restoration project site in terms of geographical and social similarities, based on 4 criteria: (a) similarity in non-market commodity, (b) similarity in the affected population, (c) similarity in property rights, and (d) similarity in ecosystem types and coastal defence measures. The benefits then were paired with the costs of the projects for the BCR calculations.

All economic costs and benefits were standardised and converted to 2021 USD values. To do so, first, we converted the monetary values reported in each individual study to the official local currency for the study year using the official exchange rates. We then used the official gross domestic product (GDP) deflator to adjust these values to the local currency values for the year 2021. Finally, all values were standardised to 2021 USD values, using the purchase power parity (PPP) conversion factors. The official exchange rates, GDP deflators, and PPP conversion factors were extracted from the World Bank's database[80].

The economic costs of coastal defence projects were calculated as 2021 USD ha$^{-1}$ for soft and hybrid measures, and 2021 USD m.m$^{-1}$ shoreline for hard measures. From the first year following the implementation of the adaptation measures, we accounted for annual operating or project maintenance costs. These costs amounted to 2.5% of the original financial capital costs for all adaptation options and for all habitats, with the exception of coral reefs that are self-sustaining following restoration (i.e. maintenance cost of 0%)[18].

The economic benefits of coastal defence projects were converted to 2021 USD ha$^{-1}$ yr$^{-1}$ for soft and hybrid measures, and to 2021 USD m.m$^{-1}$ shoreline yr$^{-1}$ for hard measures. We calculated the total benefit of coastal defence options for a period of 20 years at social discount rates of −2%, 4.5%, and 8%[18]. The negative rate −2% reflects the scenario that ecological degradation and resource depletion will deteriorate the future conditions, thus resulting in a greater value of any additional wealth[18]. The rate 8% reflects a scenario of slowing economic development and rising energy prices, thus overestimating the risk-adjusted opportunity costs[18]. The net present economic benefit was calculated by summing the average annual values of the total economic benefits provided by the individual coastal defence options. For soft and hybrid options (excluding beach and sand dune nourishment), the benefits were calculated from the fifth year following the initial development to allow enough time for the restored/planted mangroves, saltmarshes, and wetlands to reach a certain level of

maturity to provide benefits[18]. For beach nourishment and hard measures, benefits were calculated from the first year following the initial development.

The benefit-cost ratios were then calculated for 20 years using Eq. 1:

$$BCR = \frac{|PV[Benefits]|}{|PV[Costs]|} = \frac{\sum_t^T B_{total}/(1+r)^t}{C_{total} + \sum_{t=1}^T C_{management}/(1+r)^t} \quad (1)$$

where PV is the present value; t is the year of calculation (t = 1 when calculating benefits for soft engineering and hard measures and t = 5 for soft and hybrid EbAs); B is the total economic benefits; C is the total economic costs; and r is the discount rate. We calculated BCR for each of the 96 coastal defence projects spanning soft, hybrid, hard measures for three discount rates −2%, 4.5%, and 8%. To ensure the robustness of the results, we excluded from the analysis the outliers (e.g. projects with abnormally high costs or benefits). For transparency, these outliers are highlighted in the database in Supplementary Data 10.

Cost-benefit analysis has only been conducted for coastal defence options that entail human interventions (i.e. hard, hybrid, soft measures) and not for natural measures or unvegetated natural systems, given the lack of data for the investment costs of these options.

## Synthesis of findings

We synthesised the findings of the systematic review using the Strengths – Weaknesses – Opportunities – Threats (SWOT) approach. The Strengths and Weaknesses refer to the internal characteristics of the defence options themselves for coastal adaptation (e.g. performance), while the Opportunities and Threats refer to the wider system characteristics that support or hinder the design and implementation of each defence option for coastal adaptation (e.g. institutional/technological/funding circumstances)[81]. For each coastal defence option, we derive statements for each SWOT element from (a) results of the meta-analysis, (b) qualitative/quantitative findings or discussion points reported in individual studies, or (c) our critical understanding of the broader literature. The entire synthesis approach relies on our expert judgement and the source of each SWOT statement is indicated accordingly.

## Challenges and limitations

First, this study relies on the academic literature specifically peer-reviewed papers and published conference papers. Data from grey literature that did not meet quality criteria was extracted but analysed in the Supplementary Box 2. We consciously decided to omit the grey literature from the main analysis to ensure the reproducibility, reliability and quality of the meta-analysis. Thus, while our meta-analysis can indicate the current scientific evidence of the effectiveness of different coastal defence options, it should not be considered as the totality of the evidence.

Second, although our review used a broad range of possible keywords related to coastal defences, these terms were confined to reflect climate change adaptation and natural disaster risk reduction. It was not possible to include all keywords related to coastal adaptation, defences and protection against natural disasters. The reviewed papers were published in English, but studies on coastal defences may also be published in other languages such as Portuguese, French, or Mandarin, among many others. Although we believe that the search process has allowed for a very good identification of the relevant international literature on this topic, keyword selection and language restriction may introduce biases in geographical representation, and possibly the direction and magnitude of mean effect sizes.

Third, findings for some individual and aggregate functions are based on small numbers of observations (n < 3) due to the general lack of literature on specific topics. For the benefits of the reader, these

functions are indicated in asterisks in the respective figures. We strongly encourage the cautious interpretation and generalisation of findings for these comparisons. To ensure the robustness of the results, we re-calculated all relevant aggregate functions omitting individual functions with few observations ($n < 3$) (see Supplementary Table 8, Supplementary Materials). The differences in effect sizes before and after removing the functions with few observations are rather minor in terms of the direction of effect sizes, the magnitudes, and their 95% CIs. We therefore are confident that the results are robust.

Fourth, we are aware of the presence of publication biases in the reviewed studies. The Funnel plot and Egger's test are popular methods for detecting publication biases[23]. However, none of the currently available methods has the desirable statistical capacity to deal with extreme heterogeneity in true effect sizes[82]. Both the funnel plot and Egger's test have inherent limitations[83]. Although we detected publication biases in some areas of our analysis, we did not conduct further investigations to ascribe values to potentially 'missing' studies. As publication biases are unavoidable in scientific research, we strongly encourage the cautious interpretation and generalisation of our findings.

## Data availability
The data that support the findings of this study are available in Figshare with the identifier https://doi.org/10.6084/m9.figshare.22672450. The source data for plotting figures can also be found in the above link. The study quality assessment table is also available in the above link.

## Code availability
The R code used in this study is available in Figshare with the identifier https://doi.org/10.6084/m9.figshare.22672450.

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

## Acknowledgements

This study was funded partly by a Grant-in-Aid Research Fellowship for Young Scientists offered by the Japan Society for the Promotion of Science (23KJ0544) (L.H), a Grant-in-Aid for Scientific Research A offered by the Japan Society for the Promotion of Science (22H00567) (A.G.), and Singapore Ministry of Education Academic Research Fund grants MOE2019-T3-1-004 and MOET32022-0006 (A.D.S), This work is EOS contribution number 573 (A.D.S).

## Author contributions

All authors contributed intellectual input and assistance to this study. L.H. and A.G. designed the research. L.H. conducted the literature search and data extraction. L.H., W.Q., and J.S. conducted data analysis and visualization. L.H. wrote the first draft of the manuscript, and L.S., A.S., and A.G. contributed substantially to revisions.

## Competing interests

The authors declare no competing interests.
