## [Peer Review File · Nature Communications]

Meta-analysis indicates better climate adaptation and mitigation performance of hybrid engineering-natural coastal defence measuresREVIEWER COMMENTS

Reviewer #1 (Remarks to the Author):

The authors have made a commendable effort in collecting and analyzing a global dataset of studies on different types of coastal risk reduction measures. There are several challenges to robust literature-based comparisons of soft vs. hybrid vs hard defence measures that the authors describe and the approach they adopt to circumvent some of these challenges, i.e. pairwise comparisons of the performance of different measures against a 'baseline' measure of unvegetated natural systems is novel and innovative.

However, I have significant concerns regarding multiple aspects of the analysis and interpretation of the results that make this manuscript unpublishable in its current format, which I detail below.

1. The paper faces a critical challenge in terms of the large bias in scientific literature towards the assessment of soft and hybrid (e.g., "living shorelines" in the US) as suggested by the authors and as evidenced also by the list of papers cited in this manuscript. While hard structures are by far the most prominent method of coastal risk reduction to date, there are far fewer assessments of the performance of these structures within scientific literature, relative to scientific studies that focus on assessing EbA options. There is a lot more information about hard structures in grey literature and reports from different parts of the world, though these are not always peer-reviewed. I fear that the focus on scientific studies can significantly bias both the results and the interpretation of these results. I suggest the authors address this issue more explicitly and in greater detail. For this it will greatly help if the authors include information on the distribution of observations by type of measure and by type of observation (e.g. how many observations do the authors have for WR by hard; CBA for hard; WR for EbA, etc).

2. It appears that the authors exclude beaches and sand dunes from their analysis - i.e., I do not see these under the definitions for "hard", "hybrid" or unvegetated" and they are explicitly excluded from "soft" (Table S4). Beaches and sand dunes that are periodically nourished are a widespread and critical coastal risk reduction method that should be included in any comprehensive global assessment of coastal risk reduction measures (see, for example, Toimil et al., 2023 - <https://www.nature.com/articles/s41467-023-39168-z>).

3. The cost-benefit analysis approach does not seem robust to me. Cost-benefit analyses are typically intended for assessing the context-specific cost-effectiveness of a project. Dividing separate assessments of total reported benefits by total reported costs across a global collection of studies can produce very misleading results in general, and is all the more an issue considering the bias towards published studies on the costs and benefits of EbA relative to similar studies on hard structures. Also, this approach does not account for large variations in costs by region. One paper cited in this manuscript - a few studies note large variability in EbA and hard structure construction costs across different parts of the world, which has also been reported in other studies (see IPCC SROCC 2019, Chapter 4).

4. Another critical, related issue is that a comparative meta-analysis should, in my opinion, not ignore is the large context-specific variation in the baseline level of risk across different types of measures, one metric for this being the incident wave energy. For example, most studies of wave transformation across habitats in the field note very low incident wave energies on the order of a few centimeters, due to the constraints of field data collection (see Narayan et al., 2016). On the other hand coastal defense structures are typically built for conditions of higher incident wave energies than these measurements. In the authors' database on wave reduction, how was this accounted for?

Overall, while the strength of this manuscript lies in its novel approach to potentially compare the performance of different types of measures, the approach needs to be considerably revised in order to

avoid critical biases in terms of the methods and metrics by which different measures are compared and with a much more thorough consideration of biases in the number and type of studies across different types of measures.

Reviewer #2 (Remarks to the Author):

This manuscript fills an important gap in the literature in terms of doing a thorough meta-analysis of coastal defense papers examining natural, hybrid, and built options and how they compare in performance for coastal resilience, climate mitigation (carbon sequestration and storage) and cost-effectiveness. Some of the most important points are the gaps in our knowledge that remain since there are very few studies that compare different coastal defense structures but under similar storm conditions. This is a very large gap. But also the finding that all coastal defense measures (including the soft and hybrid ones) had positive economic returns is important. This helps build the case that society should be making a much larger investment in coastal defense now because it is only going to get more expensive as time goes on and this paper helps to build the case that a substantial amount of that investment would make sense to use soft or hybrid approaches because they performed as well or better than the hard measures in many of the metrics used for the analysis. I think this paper will make an important contribution to the literature. I provide the following suggestions for improvement:

1) It would be good to highlight in the abstract some of the policy implications of the results because those are the most important/noteworthy conclusions from the analysis. Like the points made in Lines 247-250 and 269-271. The fact that all of the coastal defense treatments examined had a positive economic return and therefore this really suggests investments now in coastal defense make a lot of sense seems like an important one. We need to act soon in order to prevent a lot of future harm. Perhaps just 1-2 final sentences that captures the importance of these results for coastal planning and decision making. It might also be good to highlight the gaps in our knowledge that remain as well.

2) The curing of concrete which is used in most hard shoreline projects releases a great deal of CO₂. I think it is important to include this in the discussion. For future comparison in particular, projects should estimate the CO₂ released based on the amount of concrete used and this should be added as a recommendation so that future comparisons are more accurate when comparing GHG emissions. It isn't the case that hard structures have NO GHG impacts which is what having a zero there seems to imply. They have a negative climate change impact because they emit GHG. This fact should be discussed and highlighted as relevant to the comparisons. Even though there are not numbers for climate change mitigation for hard infrastructure because they aren't sequestering carbon, those projects are likely releasing GHG and therefore have an even stronger contrast to the natural and hybrid options that are sequestering carbon therefore having a beneficial climate change impact. This should be part of the discussion to make it clear that not having numbers for the hard projects does not mean a zero value for GHG, but that there actually is likely GHG release from hard projects.

3) For the BCA analysis, Line 633: Did benefits decrease with age for hard structures due to wear and tear and needed maintenance? Explain how/if the 20 year time frame impacted the hard structure economic estimates.

4)

A few smaller edits/comments:

1) I saw a couple of papers that were not referenced but could be: Check out the work of Dr. Rachel Gittman at Eastern Carolina University: https://scholar.google.com/scholar?hl=en&as_sdt=0%2C21&q=Rachel+Gittman&btnG= I think she has more papers that could be relevant to this analysis but in particular the one that compares marshes to bulkheads during a hurricane would be useful to include in the discussion where those few papers that have done this type of analysis are discussed: <https://www.sciencedirect.com/science/article/pii/S096456911400297X> Also, the one that looks at

cost-effectiveness of hybrid options might be of use: <https://www.mdpi.com/2071-1050/10/2/523>

2) Line 25: Move significantly to right after storms

3) Lines 36-38: the sentence is very awkward. I think the end of the sentence should be modified perhaps by putting a colon and then saying "; this implies/suggests a strong need for effective coastal defence in order to keep pace with accelerated climate change impacts."

4) Line 64: I would change would to "are"

5) Lines 148-151: These comparisons are true. Hard structures are designed for these specific measures and really these alone. So, somehow this comparison almost seems slanted in favor of the hard structures. Might be worth adding a sentence to the effect of "These results confirm what we know, that the hard structures are designed very specifically for risk reduction and therefore perform well for these functions despite the fact that they do not perform well for other desired functions like climate mitigation."

6) Line 153: Might be worth mentioning/discussing somewhere that non-vegetated tidal areas serve a different role in the landscape from vegetated tidal areas and that affects their risk reduction function. It is not a surprise to me that they do not function as well for risk reduction as the other options.

7) Line 2212: I would say "hard measures better at:"

8) Line 220: similarly, need the ly

9) Line 242: don't need the "ing" on protecting...

10) Line 275, I would start the sentence with "Of note" or "Additionally" because it isn't really a however kind of transition.

11) Line 276: I would add at the end of the sentence, "which would make the benefit calculations increase for soft and natural options."

12) Line 287. Consider adding a last sentence like "But in cases where natural options have been destroyed or degraded, soft or hybrid options are the next best option."

13) Line 340: Begin with "Additionally,"

14) Line 539: No s on denotes

15) Table 1: Different discount rates are mentioned in the text when referring to Table 1 but I don't see how those discount rates apply to Table 1. Should be explained.

16) Table 2: Soft measures weaknesses box: 2nd bullet should be "dependent" and last bullet it is not clear what "distinct human interactions during implementation means" and this was used again in the hybrid measures weakness box. Please explain this somewhere.

17) Figure 3: Clarify the difference between "wetland" and "marsh."

RESPONSE TO REVIEWERS

Reviewer #1 (Remarks to the Author):

The authors have made a commendable effort in collecting and analyzing a global dataset of studies on different types of coastal risk reduction measures. There are several challenges to robust literature-based comparisons of soft vs. hybrid vs hard defence measures that the authors describe and the approach they adopt to circumvent some of these challenges, i.e. pairwise comparisons of the performance of different measures against a 'baseline' measure of unvegetated natural systems is novel and innovative.

However, I have significant concerns regarding multiple aspects of the analysis and interpretation of the results that make this manuscript unpublishable in its current format, which I detail below.

Thanks for the feedback. We revised the paper following all your comments.

1. The paper faces a critical challenge in terms of the large bias in scientific literature towards the assessment of soft and hybrid (e.g., "living shorelines" in the US) as suggested by the authors and as evidenced also by the list of papers cited in this manuscript. While hard structures are by far the most prominent method of coastal risk reduction to date, there are far fewer assessments of the performance of these structures within scientific literature, relative to scientific studies that focus on assessing EbA options. There is a lot more information about hard structures in grey literature and reports from different parts of the world, though these are not always peer-reviewed. I fear that the focus on scientific studies can significantly bias both the results and the interpretation of these results. I suggest the authors address this issue more explicitly and in greater detail. For this it will greatly help if the authors include information on the distribution of observations by type of measure and by type of observation (e.g. how many observations do the authors have for WR by hard; CBA for hard; WR for EbA, etc).

We acknowledge the possibility of bias of our results due to the reason mentioned by the Reviewer. The literature search in the original manuscript (conducted in May 2022) used Elsevier Scopus and ISI Web of Science Core Collection and was limited to peer-reviewed academic papers. In the original manuscript, we consciously decided to omit the grey literature to ensure the reproducibility, reliability and the quality of our analysis. However, as pointed by the reviewer, this search produced a relatively smaller number of peer-reviewed papers on hard structures compared to soft and hybrid measures. We now include in the analysis grey literature on hard measures that passes certain quality control criteria. We have taken specific steps to tackle this issue as follows:

First, we conduct an additional literature search in July 2023 during the revision. For the peer-reviewed literature we used Elsevier Scopus and ISI Web of Science Core Collection and for the grey literature we use BASE database. This second search contains a more comprehensive set of keywords to capture information for more coastal defence measures (i.e. include beach and sand dune to also address comment 2 of Reviewer 1).

We need to point that all papers included in the meta-analysis must contain paired experiments comparing the performance of at least one coastal defence option (i.e., soft, hybrid or hard) with another defence option or a baseline (e.g. natural measures or unvegetated systems). Studies that provide performance of a single coastal defence option without a comparison are not eligible, whether they report on EbAs or hard measures.

For the grey literature, in the main meta-analysis reported in the manuscript we consciously decided to include only conference proceeding papers and doctoral dissertations on the performance of hard measures. We then conducted a careful quality critical appraisal on the identified grey literature for internal validity including research aims and objectives, data collection, data analysis, results and conclusions following the guideline and checklist (see Methods). Following this quality check an additional 4 conference proceeding papers on hard structures were included in the revised analysis. Data from this grey literature was then extracted and analysed in the meta-analysis reported in the manuscript.

For other types of grey literature such as consultancy reports, governmental reports, and reports to funders, we have significant concern over the possible quality issues. Often such documents report the outcomes of coastal defence projects to whom the organisations have received funding to undertake the projects or oppose it. Furthermore, they also fail several of the quality criteria used to judge the quality of information, which is a critical aspect of meta-analyses. However, we acknowledge that data from these reports are also valuable, so we extracted and analysed it in the Supplementary Material. We believe that including this data in the Supplementary Materials can provide additional judgement of whether the effect sizes are different between peer-reviewed/conference papers and grey literature regarding the performance of hard measures, while not compromising the robustness and reliability of the meta-analysis. After literature search and full-text screening, we identified 9 reports including 27 observations of pair-wise comparison between hard structures and control options (see database, worksheet: grey literature). Effect sizes were calculated and reported in the Supplementary Materials (see Box S2).

Effect sizes between peer-reviewed/conference papers and grey literature were compared. Regarding the wave attenuation function, the effect sizes are higher in the grey literature (SMD=19.52, 95% CIs = 6.16-33.44, n=26) compared to the peer-reviewed literature (SMD=6.28, 95% CIs=2.78-9.80, n=9). Despite these differences in the magnitude of effect sizes, the pool effect sizes show a similar direction of the sign (positive sign, 95% CIs do not intersect with zero). We then conclude that the results of the meta-analysis of the peer-reviewed papers, although lower in effect sizes, also reflect similar findings to what has been reported in the grey literature.

To sum up, in this revised manuscript, we (1) redid the meta-analysis and revised all Figures accordingly, (2) included information on the number of observations by type of measure and by type of functions between the comparison groups in both Figures and main text in the Results section (as recommended by the reviewer), (3) highlighted the areas that needed to be interpreted with caution due to the lack of evidence in the current literature (e.g. see lines 131-133, 159-161), (4) included a paragraph in the Discussion (lines 325-329) about the differences in the evidence reported in the peer-reviewed/conference papers and grey literature on the performance of hard measures, and (5) included a paragraph on the possible bias towards EbAs due to the overwhelming amount of EbA literature compared to hard measures in the reviewed studies (lines 318-332).

We hope the above approach is reasonable and inspires more confidence on the rigor of our manuscript.

2. It appears that the authors exclude beaches and sand dunes from their analysis - i.e., I do not see these under the definitions for "hard", "hybrid" or unvegetated" and they are explicitly excluded from "soft" (Table S4). Beaches and sand dunes that are periodically nourished are a widespread and critical coastal risk reduction method that should be included in any comprehensive global assessment of coastal risk reduction measures (see, for example, Toimil et al., 2023 - <https://www.nature.com/articles/s41467-023-39168-z>).

In the original manuscript, we had excluded beach and sand dunes from the analysis as we wanted to focus on ecosystem-related adaptation from soft and hybrid measures. We now include beaches and sand dunes in the meta-analysis as an important coastal defence measure. However, the inclusion of the literature needs to be done carefully to ensure proper analysis, while it also necessitates the slight revision of the terminology used in the paper.

In the revised manuscript, following the addition of beaches/dunes we changed the term ‘ecosystem-based adaptations’ (EbAs) to ‘nature-based solutions’ (NbS) and included beaches and sand dunes as a subgroup in soft and hybrid NbS coastal defence measures, alongside other marine ecosystems such as mangroves, coral reefs, and saltmarshes. Definitions of the comparison groups were revised to include beaches and sand dunes, as follows, if beaches and sand dunes:

- are actively managed and nourished for coastal defence (i.e. beach nourishment, sand dune planting), they fall in the group “**soft measures**”.

- are actively managed and nourished for coastal defence (i.e. beach nourishment, sand dune planting) and combined with hard built structures, they fall in the group “**hybrid measures**”.

- do not include active management or nourishment for coastal defence, they fall in the group “**unvegetated natural systems**”.

For the additional search (as outlined in response to Comment 1) on beach and sand dune-related coastal defence measures, we searched the database using the keywords: beach* OR “sand dune*” AND (nourish* OR plant* OR advance* OR artificial OR creat* OR regener* OR creat* OR hybrid OR recover)

After full text screening and quality control, 13 studies on beach and sand dune projects were identified and included in the meta-analysis. We included beaches and sand dunes as a subgroup of soft and hybrid coastal defence measures following the distinctions made above and conducted meta-analysis for both risk reduction functions and cost-benefit analysis.

3. The cost-benefit analysis approach does not seem robust to me. Cost-benefit analyses are typically intended for assessing the context-specific cost-effectiveness of a project. Dividing separate assessments of total reported benefits by total reported costs across a global collection of studies can produce very misleading results in general, and is all the more an issue considering the bias towards published studies on the costs and benefits of EbA relative to similar studies on hard structures. Also, this approach does not account for large variations in costs by region. One paper cited in this manuscript - a few studies note large variability in EbA and hard structure construction costs across different parts of the world, which has also been reported in other studies (see IPCC SROCC 2019, Chapter 4).

We appreciate this point. In the original manuscript, we followed a well-established cost-benefit methodological procedure proposed by The Economics of Ecosystem and Biodiversity (TEEB, 2010). This approach has been a standard economic evaluation of ecosystem services and cost-benefit analysis of ecosystem restoration projects, adopted in many large-scale exercises from global assessment reports such as the IPBES value assessments (IPBES, 2023) as well as other high quality academic papers (e.g. Richardson *et al.*, 2015; Chen, 2020; Stewart-Sinclair *et al.*, 2021).

However, we agree about the inherent limitation of this method in how it cannot consider well the context-specific variation in cost and benefit values of the coastal defence projects globally. In the revised manuscript, we conducted cost-benefit analysis for each coastal defence project instead of combined costs and benefits across a collection of studies. Overall, 96 single coastal defence projects with pairwise cost

and benefit values were extracted from the peer-reviewed literature, including 55 observations of soft, 19 observations of hybrid, and 24 observations of hard projects.

Among, these 96 projects, 50 projects reported pair-wise information on the total costs and total benefits in the same sites within the same projects and 46 projects (all soft and hybrid projects) only reported the costs. For the missing information on the total benefits of these 46 projects, we used the value transfer approach to pair benefit values with the cost values. Benefit transfer is a well-established approach that has been used widely in academic literature on economic evaluation (Richardson *et al.*, 2015; Himes-Cornell, Grose and Pendleton, 2018; Stewart-Sinclair *et al.*, 2021). More details about the adopted value transfer method are included in the Methodology section of the revised manuscript.

All cost and benefit values were standardised and converted to 2021 USD. We then calculated benefit-cost ratio (BCR) for each project under 3 scenarios of discount rates -2%, 4.5%, and 8%. We used box plots to illustrate the BCRs of these 96 projects.

We hope that this approach now inspires more confidence in the robustness of the cost-benefit analysis.

4. Another critical, related issue is that a comparative meta-analysis should, in my opinion, not ignore is the large context-specific variation in the baseline level of risk across different types of measures, one metric for this being the incident wave energy. For example, most studies of wave transformation across habitats in the field note very low incident wave energies on the order of a few centimeters, due to the constraints of field data collection (see Narayan *et al.*, 2016). On the other hand coastal defense structures are typically built for conditions of higher incident wave energies than these measurements. In the authors' database on wave reduction, how was this accounted for?

We agree that comparative meta-analysis should not ignore the large context-specific variation in the baseline level of risk across studies. Although we did briefly mention the level of risks in the reviewed studies in the Discussion part in the original manuscript (i.e. most of the studies conducted data collection at relatively low-risk contexts, which has been one of the major limitations of the current literature) (see lines 309-317 of the revised version), we did not address this issue comprehensively in the meta-analysis.

In the revised manuscript, we conduct an additional subgroup analysis on the performance of coastal defence measures in different risk contexts and energy levels. Specific steps outlined below were taken:

First, we extracted information related to morphodynamic characteristics in each study. Data on significant wave height and storm conditions are used as indicators to classify the contextual risk condition into high-energy and low-energy conditions (see Methods).

Second, subgroup analysis was conducted to compare the performance of different coastal defence measures in low-energy and high-energy conditions. Results are reported in Figure 3.

Third, we include a paragraph in the Discussion section about the results and indicated a significant gap in the current literature on the lack of pairwise experiment setup in extreme weather events, high-risk and high-energy contexts (see lines 290-317 of the revised version).

Following the revisions for Comments 1-2 and to avoid confusing the reader, we report in the paper only the results of the second literature search in July 2023. This basically shows the state of literature as of July 2023. For transparency purposes we provide below the PRISMA figure for both literature searches for the reviewer's information.

Reviewer #2 (Remarks to the Author):

This manuscript fills an important gap in the literature in terms of doing a thorough meta-analysis of coastal defense papers examining natural, hybrid, and built options and how they compare in performance for coastal resilience, climate mitigation (carbon sequestration and storage) and cost-effectiveness. Some of the most important points are the gaps in our knowledge that remain since there are very few studies that compare different coastal defense structures but under similar storm conditions. This is a very large gap. But also the finding that all coastal defense measures (including the soft and hybrid ones) had positive economic returns is important. This helps build the case that society should be making a much larger investment in coastal defense now because it is only going to get more expensive as time goes on and this paper helps to build the case that a substantial amount of that investment would make sense to use soft or hybrid approaches because they performed as well or better than the hard measures in many of the metrics used for the analysis. I think this paper will make an important contribution to the literature. I provide the following suggestions for improvement:

Thanks for the positive feedback. We revised following all your comments.

1) It would be good to highlight in the abstract some of the policy implications of the results because those

are the most important/noteworthy conclusions from the analysis. Like the points made in Lines 247-250 and 269-271. The fact that all of the coastal defense treatments examined had a positive economic return and therefore this really suggests investments now in coastal defense make a lot of sense seems like an important one. We need to act soon in order to prevent a lot of future harm. Perhaps just 1-2 final sentences that captures the importance of these results for coastal planning and decision making. It might also be good to highlight the gaps in our knowledge that remain as well.

We revised the abstract and introduction taking on board your comment. The policy implications have been highlighted in lines 19-22.

2) The curing of concrete which is used in most hard shoreline projects releases a great deal of CO₂. I think it is important to include this in the discussion. For future comparison in particular, projects should estimate the CO₂ released based on the amount of concrete used and this should be added as a recommendation so that future comparisons are more accurate when comparing GHG emissions. It isn't the case that hard structures have NO GHG impacts which is what having a zero there seems to imply. They have a negative climate change impact because they emit GHG. This fact should be discussed and highlighted as relevant to the comparisons. Even though there are not numbers for climate change mitigation for hard infrastructure because they aren't sequestering carbon, those projects are likely releasing GHG and therefore have an even stronger contrast to the natural and hybrid options that are sequestering carbon therefore having a beneficial climate change impact. This should be part of the discussion to make it clear that not having numbers for the hard projects does not mean a zero value for GHG, but that there actually is likely GHG release from hard projects.

Thanks for pointing this out. We added a paragraph in the Discussion related to the CO₂ emission of hard measures (Line 275-281). Also reflecting the request of Reviewer 1, we have conducted extra literature review on carbon emissions of hard measures and reported these in Box S2 in Supplementary Materials.

3) For the BCA analysis, Line 633: Did benefits decrease with age for hard structures due to wear and tear and needed maintenance? Explain how/if the 20 year time frame impacted the hard structure economic estimates.

We did not account for the decrease in the benefits with age for hard structures but instead we account for the increase in maintenance costs over time. This is common for cost-benefit analysis of hard projects in the literature (see de Ruig *et al.*, 2019; Cupać, Trbić and Zahirović, 2020). As the timeline for most of the hard structures (and hybrid projects) is more than 30 years, the chosen timeline of 20 years may underestimate some of the economic returns of these measures. We discussed this issue in the original manuscript (lines 343-349, which remains unchanged in the revised manuscript). We opted for a 20-year period as this timeline is a standard practice for ecosystem-related projects following TEEB guidelines, hence serving the purpose of comparison between hard, soft, and hybrid measures.

We have noted in the Discussion that our CBA may underestimate the economic return for all coastal defence measures (i.e. lack of inclusion of intangible benefits and cultural ecosystem services of soft and hybrid measures, using a 20-year period for hard and hybrid measures). The key point here is, despite the possibility of underestimating the BCRs in our analysis, the results show a positive economic return for all soft, hybrid, and hard measures. We believe our results can make a case for the investment in coastal disaster risk reduction globally.

4) A few smaller edits/comments:

1. I saw a couple of papers that were not referenced but could be: Check out the work of Dr. Rachel Gittman at Eastern Carolina University: https://scholar.google.com/scholar?hl=en&as_sdt=0%2C21&q=Rachel+Gittman&btnG=

I think she has more papers that could be relevant to this analysis but in particular the one that compares marshes to bulkheads during a hurricane would be useful to include in the discussion where those few papers that have done this type of analysis are discussed: <https://www.sciencedirect.com/science/article/pii/S096456911400297X> Also, the one that looks at cost-effectiveness of hybrid options might be of use: <https://www.mdpi.com/2071-1050/10/2/523>

Thanks for the recommendation. We added these papers in the Discussion.

2. Line 25: Move significantly to right after storms.

The statement was revised.

3. Lines 36-38: the sentence is very awkward. I think the end of the sentence should be modified perhaps by putting a colon and then saying “; this implies/suggests a strong need for effective coastal defence in order to keep pace with accelerated climate change impacts.”

The statement was revised.

4. Line 64: I would change would to “are”

The statement was revised.

5. Lines 148-151: These comparisons are true. Hard structures are designed for these specific measures and really these alone. So, somehow this comparison almost seems slanted in favor of the hard structures. Might be worth adding a sentence to the effect of “These results confirm what we know, that the hard structures are designed very specifically for risk reduction and therefore perform well for these functions despite the fact that they do not perform well for other desired functions like climate mitigation.”

The statement was added.

6. Line 153: Might be worth mentioning/discussing somewhere that non-vegetated tidal areas serve a different role in the landscape from vegetated tidal areas and that affects their risk reduction function. It is not a surprise to me that they do not function as well for risk reduction as the other options.

We added the statements in the Methodology in the definition of control groups and treatment groups.

7. Line 2212: I would say “hard measures better at:”

The statement was revised.

8. Line 220: similarly, need the ly

The statement was revised.

9. Line 242: don't need the "ing" on protecting...

The statement was revised.

10. Line 275, I would start the sentence with "Of note" or "Additionally" because it isn't really a however kind of transition.

The statement was revised.

11. Line 276: I would add at the end of the sentence, 'which would make the benefit calculations increase for soft and natural options.'

The statement was revised.

12. Line 287. Consider adding a last sentence like "But in cases where natural options have been destroyed or degraded, soft or hybrid options are the next best option."

The statement was added.

13. Line 340: Begin with "Additionally,"

The statement was revised.

14. Line 539: No s on denotes\

The statement was revised.

15. Table 1: Different discount rates are mentioned in the text when referring to Table 1 but I don't see how those discount rates apply to Table 1. Should be explained.

We removed Table 1 as we changed the cost-benefit analysis approach following the comment of Reviewer 1.

16. Table 2: Soft measures weaknesses box: 2nd bullet should be "dependent" and last bullet it is not clear what "distinct human interactions during implementation means" and this was used again in the hybrid measures weakness box. Please explain this somewhere.

The statement was revised.

17. Figure 3: Clarify the difference between "wetland" and "marsh."

We changed 'wetland' to 'unspecified wetland'.

Reference

Chen, H. (2020) 'Complementing conventional environmental impact assessments of tourism with ecosystem service valuation: A case study of the Wulingyuan Scenic Area, China', *Ecosystem Services*. Elsevier B.V., 43. doi: 10.1016/j.ecoser.2020.101100.

Cupać, R., Trbić, G. and Zahirović, E. (2020) 'Cost-benefit analysis of climate change adaptation measures in Bosnia and Herzegovina', *Euro-Mediterranean Journal for Environmental Integration*. Springer International Publishing, 5(2), pp. 1-9. doi: 10.1007/s41207-020-00160-4.

Himes-Cornell, A., Grose, S. O. and Pendleton, L. (2018) 'Mangrove ecosystem service values and methodological approaches to valuation: Where do we stand?', *Frontiers in Marine Science*, 5(OCT), pp. 1–15. doi: 10.3389/fmars.2018.00376.

IPBES (2023) *Policy support tool: Value transfer method*. Available at: <https://www.ipbes.net/policy-support/tools-instruments/value-transfer-method>.

Richardson, L. *et al.* (2015) 'The role of benefit transfer in ecosystem service valuation', *Ecological Economics*. Elsevier B.V., 115, pp. 51–58. doi: 10.1016/j.ecolecon.2014.02.018.

de Ruig, L. T. *et al.* (2019) 'An economic evaluation of adaptation pathways in coastal mega cities: An illustration for Los Angeles', *Science of the Total Environment*. The Authors, 678, pp. 647–659. doi: 10.1016/j.scitotenv.2019.04.308.

Stewart-Sinclair, P. J. *et al.* (2021) 'Spatial cost–benefit analysis of blue restoration and factors driving net benefits globally', *Conservation Biology*, 35(6), pp. 1850–1860. doi: 10.1111/cobi.13742.

TEEB (2010) 'The Economics of Ecosystems and Biodiversity: Ecological and Economic Foundations, edited by Pushpam Kumar, 2010, London and Washington: Earthscan, ISBN 978-1-84971-212-5 (HB) Price £49.99 [Earthscan have offered a 20% discount off the book for EDE reader', *Environment and Development Economics*, 16(2), pp. 239–242. doi: 10.1017/s1355770x11000088.

REVIEWER COMMENTS

Reviewer #1 (Remarks to the Author):

The authors have made substantial revisions to the manuscript to address the concerns raised in the first reviews. The revisions to the B-C analyses and the inclusion of low v high wave energy environments are very good to see. I still have some concerns that I think need to be addressed before the manuscript is publishable. I have listed these point-wise below:

1. My biggest remaining concern is that of sample sizes. Most of the parameters have an $n < 20$. The authors should clarify if they use the small n correction for hedge's g (SMD) for small samples, typically applied to all $n < 50$. If not, these calculations should be modified using this correction (see <https://www.itl.nist.gov/div898/software/dataplot/refman2/auxillar/hedgeg.htm>). Related to this correction, in my opinion, observations with an $n < 3$ should not be reported. For such small numbers, effect size and CI statistics can be misleading especially when these are being used to interpret a general direction in trends, which is then used as a foundation for Figure 5, and the accompanying Results and Discussions.

2. It is good to see the division between high and low wave energy sites. The authors should make this division clear from the start in Figure 1 (i.e. have two rows for wave attenuation in Figure 1; one for low energy sites and one for high energy sites). The reasons for this are twofold: 1) Figure 1 is still potentially misleading, in that wave attenuation performance and variation in this performance are not separated out by whether these are occurring under low or high wave energy environments. 2) when discussing wave attenuation as a risk reduction function, and not just as a metric of its own, initial wave height is crucial (at very low wave heights, risk is very low and may not be a concern).

3. Figure 5 should be revised following the revisions in the effect sizes for small n values described above.

4. The authors should explain the process by which Table 1 is derived in their Methods. Right now it is unclear if (and how) these lessons emerge from the meta-analysis or from other portions of the literature reviewed.

5. I question the usefulness of Figure 6 (currently wrongly labelled Figure 5), in light of the revised results. It is unclear how to interpret the two X-axis dimensions in the figure - why does high risk reduction automatically imply high cost-effectiveness? Also, what is the process by which the authors place all adaptation options as equally "High" for risk reduction and likewise, equally "High" for cost-effectiveness? The authors should discuss this in the context of previous global and smaller-scale studies that find differences in risk reduction potential as well as in cost-effectiveness across measures.

Some other minor comments:

1. In Figs 1-3, what is the difference between (0,0) and N/A?

2. The difference in x-axis limits between panels, especially in the same row, is confusing when interpreting the figures; please homogenize the x-axes for all panels in a row, or at a minimum, group panels based on similarity in their x-axis limits (e.g. in Fig 3, WA plots can be grouped together).

3. In the figure caption for Fig 1, or in Methods, please explain how the numbers for panels g-h, Hybrid v soft and Hybrid v Hard are obtained.

4. In the figure captions for Figs 1-3, and/or in Results, the authors should include an explanation of what the different SMD values mean, as these vary significantly for each panel.

5. Please indicate the n for each plot box in Fig 4a and 4b.

Reviewer #2 (Remarks to the Author):

The authors have addressed my concerns and the revisions and new analysis have greatly improved the manuscript.

I found one typo. Line 232 where relativity should be relatively.

REVIEWER COMMENTS

Reviewer #1 (Remarks to the Author):

The authors have made substantial revisions to the manuscript to address the concerns raised in the first reviews. The revisions to the B-C analyses and the inclusion of low v high wave energy environments are very good to see. I still have some concerns that I think need to be addressed before the manuscript is publishable. I have listed these point-wise below:

Thank you for your positive feedback. We revised following your comments.

1. My biggest remaining concern is that of sample sizes. Most of the parameters have an $n < 20$. The authors should clarify if they use the small n correction for hedge's g (SMD) for small samples, typically applied to all $n < 50$. If not, these calculations should be modified using this correction (see <https://www.itl.nist.gov/div898/software/dataplot/refman2/auxillar/hedgeg.htm>). Related to this correction, in my opinion, observations with an $n < 3$ should not be reported. For such small numbers, effect size and CI statistics can be misleading especially when these are being used to interpret a general direction in trends, which is then used as a foundation for Figure 5, and the accompanying Results and Discussions.

Thanks for these two very valid concerns.

Regarding the first point, we have estimated SMD using the 'escal' function of R 'metafor' package which is appropriate for small sample size bias corrected Hedges' g (see Package Metafor R 2023). The bias corrected Hedges' g is generally preferred to other SMD estimators as it has small sample size correction and is not affected by unequal sampling variance between paired groups (see (Koricheva, Gurevitch and Mengersen, 2013)). Hence, we believe that there is no need to modify our calculations.

- For the benefit of the readers, we include a sentence explaining the Hedges' g and its small sample size correction property, with appropriate citations (see line 604-607) and that the specific R package is appropriate for this estimation (Line 678-679).

Regarding the second point on observations with $n < 3$, we understand the concern of the Reviewer but we partly agree with the comment. We will answer the two components separately.

On the one hand, we respectfully disagree about the omission of observations with $n < 3$ from the paper. One of the purposes of meta-analyses and systematic literature reviews is to provide a good grasp of what has been extensively and robustly studied, what aspects have been understudied, and identify possible directions for future research. In this sense, we think that observations with an $n < 3$ should also be reported, but with clear indications of the need for cautious interpretation and generalisation due to the small number of observations. With this in mind, we revised several parts of the manuscript as follows:

- In Figure 1, 2, and 5, we added '#' next to observations with $n < 3$ to indicate that they would require cautious interpretation due to the small number of observations.

- Improved the main text in the Results section to indicate more clearly which results would need cautious interpretation and generalisation due to the small number of observations (see line 132-134, 149-151, and 167-168).
- Included a paragraph in the limitations in the Methods about findings based on small numbers of observations (see Line 772-779)

On the other hand, we totally agree with the concern of the possible effects of individual functions with few observations on aggregate functions. For this reason, we have re-calculated all **relevant** aggregate functions omitting individual functions with few observations (n<3). Please see table below:

Comparison	Aggregate Function	Figure	Original estimates			Removed individual functions	Adjusted estimates		
			SMD	Low	High		SMD	Low	High
Soft vs. natural	Risk reduction	Figure 1a	1.7311	0.1266	3.3355	2	1.597	0.1106	3.3045
	Overall performance	Figure 1a	0.2547	-0.2747	0.7842	2	0.2250	-0.3101	0.7601
Hybrid vs. natural	Risk reduction	Figure 1b	2.6576	-0.4482	5.7635	1	2.6209	-0.7830	6.0247
	Overall performance	Figure 1b	1.2212	-1.0676	3.5099	1	1.113	-1.3041	3.5301
Hard vs. natural	Risk reduction	Figure 1c	-2.2604	-6.4284	1.9076	1	-0.345	-2.2292	1.5392
	Overall performance	Figure 1c	-2.2604	-6.4284	1.9076	1	-0.345	-2.2292	1.5392
Hybrid vs. unvegetated	Risk reduction	Figure 1e	6.3589	2.7801	9.9378	2	7.0968	2.8611	11.3325
	Overall performance	Figure 1e	5.8853	2.4971	9.2736	3	7.0968	2.8611	11.3325

We see that in most cases the adjusted SMD is only slightly affected in terms of size (in Fig 1), not affecting the colour of any arrow in Figure 4. What is more important for meta-analyses such as the one conducted in this study is that there is no change for any aggregate function on intersection with zero. Considering both points above we prefer to keep the original results in the main text. However, to reflect the concern of the Reviewer:

- We add in the Supplementary Material the Table above;
- We add clear indications in the main text about: (a) the possible effect of individual functions with few observations and large variations on aggregate functions (Line 775-779), and (b) that we do not observe major effects on the results cross-referencing where needed Table S8 from the Supplementary Material (Line 775-779).

We hope this approach addresses the Reviewer's concern.

2. It is good to see the division between high and low wave energy sites. The authors should make this division clear from the start in Figure 1 (i.e. have two rows for wave attenuation in Figure 1; one for low energy sites and one for high energy sites). The reasons for this are twofold: 1) Figure 1 is still potentially misleading, in that wave attenuation performance and variation in this performance are not separated out by whether these are occurring under low or high wave energy environments. 2) when discussing wave attenuation as a risk reduction function, and not just as a metric of its own, initial wave height is crucial (at very low wave heights, risk is very low and may not be a concern)

Thank you for your suggestion. Figure 1 has been revised accordingly. We included the low/high energy calculations for wave attenuation function in Figure 1 and in the related main text in the Result section. To avoid repetition, we omitted subgroup analysis for high/low energy in the later part and brought the results of the baseline energy level sub-group analysis earlier in the manuscript.

3. Figure 5 should be revised following the revisions in the effect sizes for small n values described above.

As there are no modifications for the effect sizes or major effects for aggregate functions (see Response 1), there is no change for Figure 5 (currently labelled Figure 4 in the revised manuscript). We added '#' to the observations with $n < 3$ to emphasise careful interpretation and generalisation of the findings in Figure 4.

Furthermore no colour changes are needed in any of the aggregated function arrows, considering the recalculation with omission of individual functions with few observations (see response and Table in Comment 1).

4. The authors should explain the process by which Table 1 is derived in their Methods. Right now it is unclear if (and how) these lessons emerge from the meta-analysis or from other portions of the literature reviewed.

Thank you for your suggestion.

First, we added a sub-section in the Methods about the approach we used to develop Table 1 (line 747-756).

Second, we revised Table 1 by adding (a) indications about the origin of each statement within the cells, and (b) a footnote explaining point (a).

5. I question the usefulness of Figure 6 (currently wrongly labelled Figure 5), in light of the revised results. It is unclear how to interpret the two X-axis dimensions in the figure - why does high risk reduction automatically imply high cost-effectiveness? Also, what is the process by which the authors place all adaptation options as equally "High" for risk reduction and likewise, equally "High" for cost-effectiveness? The authors should discuss this in the context of previous global and smaller-scale studies that find differences in risk reduction potential as well as in cost-effectiveness across measures.

We see the Reviewer's point and thanks for pointing it out. Following the Reviewer's comment we experimented with different visualisations, but we did not come up with a satisfactory one. For this reason, we decided to omit Figure 6. We believe that the narrative in the specific section conveys very well what we wanted to convey with the figure.

Some other minor comments:

1. In Figs 1-3, what is the difference between (0,0) and N/A?

(0,0) means that there are no studies that have reported findings for this function. 'N/A' means that the specific function were not amenable for comparisons between coastal defence options. For example, the carbon storage function is only provided by nature-based coastal defences that have a vegetation component (and can thus store carbon), and is not provided by hard structure.

2. The difference in x-axis limits between panels, especially in the same row, is confusing when interpreting the figures; please homogenize the x-axes for all panels in a row, or at a minimum, group panels based on similarity in their x-axis limits (e.g. in Fig 3, WA plots can be grouped together).

We revised the Figures to our best ability to homogenise the x-axes for all panels. However, as the SMD has a significantly wide range (from -14.70 to +29.71), it is not visually beneficial to have the same scale for the x-axes of all panels.

For the specific example, we have now omitted Figure 3 as it was incorporated into Figure 1 (as suggested in Comment 2 above).

3. In the figure caption for Fig 1, or in Methods, please explain how the numbers for panels g-h, Hybrid v soft and Hybrid v Hard are obtained.

The explanation was already provided in the previous manuscript in the Methods section (Line 643-653). For the benefit of the reader, we added a brief explanation in the caption of Figure 1.

4. In the figure captions for Figs 1-3, and/or in Results, the authors should include an explanation of what the different SMD values mean, as these vary significantly for each panel.

We added the explanation in the captions.

5. Please indicate the n for each plot box in Fig 4a and 4b.

We added n in the Figures.

Reviewer #2 (Remarks to the Author):

The authors have addressed my concerns and the revisions and new analysis have greatly improved the manuscript.

I found one typo. Line 232 where relativity should be relatively.

Thank you very much for your positive feedback throughout the review process. We revised the typo.

Reference

Koricheva, J., Gurevitch, J. and Mengersen, K. (2013) *Handbook of Meta-analysis in ecology and evolution*. Princeton University Press.

Viechtbauer W. (2023). Metafor: *Meta-Analysis Package for R*. Available at: <https://www.metafor-project.org>

REVIEWERS' COMMENTS

Reviewer #1 (Remarks to the Author):

Thank you to the authors for their patience with this dialogue and their sincere and thorough efforts to resolve the comments I raised. I only have two very minor comments remaining.

1. I suggest the authors add the word "and mitigation" to the title since half the figures and results focus on climate mitigation-relevant issues like carbon storage.

2. I request a minor revision to some sentences in the Results such as the following:

"Hard measures perform worse when compared to natural measures for wave attenuation both for high wave energy conditions, such as during storms or significant wave height >1m (SMD=-9.22, n=1) and low wave energy conditions (SMD=-0.97, 95% CIs=-1.84 - -0.09, n=5). " Briefly, I suggest that for the high wave energy portion of this sentence, the authors add the same explicit phrase they use for hybrid measures later on: "(note that this comparison is based on only one observation and should be interpreted cautiously)", and add a suitably modified version of this phrase for all sentences in results and discussions where $n < 3$.

I still believe that an $n=1$ does not allow for an absolute statement about the relative performance of two classes in general. However, I understand the authors' argument that they are limited by the number of studies available and agree that it is important to highlight that this is understudied in the scientific literature, but I think that this caveat should be acknowledged with a more explicit statement of caution as suggested above.

Reviewer #1 (Remarks on code availability):

I get an error message:
DOI NOT FOUND
10.6084/m9.figshare.22672450

REVIEWER COMMENTS

Reviewer #1 (Remarks to the Author):

Thank you to the authors for their patience with this dialogue and their sincere and thorough efforts to resolve the comments I raised. I only have two very minor comments remaining.

We sincerely thank the Reviewer for providing comprehensive and invaluable feedback. The manuscript has been improved substantially in its overall quality and clarity after the review process. We truly appreciate the dedication and expertise demonstrated by the Reviewer in contributing to the revision of our work.

We revise it further according to your comments as follows.

1. I suggest the authors add the word "and mitigation" to the title since half the figures and results focus on climate mitigation-relevant issues like carbon storage.

The title has been changed to incorporate 'mitigation'. The new title is 'Meta-analysis indicates better climate adaptation and mitigation performance of hybrid engineering-natural coastal defence measures.'

2. I request a minor revision to some sentences in the Results such as the following:

"Hard measures perform worse when compared to natural measures for wave attenuation both for high wave energy conditions, such as during storms or significant wave height >1m (SMD=-9.22, n=1) and low wave energy conditions (SMD=-0.97, 95% CIs=-1.84 - -0.09, n=5)" Briefly, I suggest that for the high wave energy portion of this sentence, the authors add the same explicit phrase they use for hybrid measures later on: "(note that this comparison is based on only one observation and should be interpreted cautiously)", and add a suitably modified version of this phrase for all sentences in results and discussions where $n < 3$.

I still believe that an $n=1$ does not allow for an absolute statement about the relative performance of two classes in general. However, I understand the authors' argument that they are limited by the number of studies available and agree that it is important to highlight that this is understudied in the scientific literature, but I think that this caveat should be acknowledged with a more explicit statement of caution as suggested above.

Thank you for the comment. We agree with the Reviewer that observations with $n=1$ should not be stated in an absolute statement about the relative performance of the two groups in the main text. We therefore remove this part of the sentence related to observation with $n=1$ in the main text. However we retain it in Figure 1, 2, and 4, which is indicated with the “#” symbol that denotes caution when generalising.

We also read again the Results section to make sure that all sentences/statements with few observations are also indicated properly.

Reviewer #1 (Remarks on code availability):

I get an error message:

DOI NOT FOUND: 10.6084/m9.figshare.22672450

We copied the DOI of the Figshare link provided in Nature Communications online submission platform.
We believe the link will be made for open access after the paper gets published.